# Efficient Neural Music Generation

**Max W. Y. Lam, Qiao Tian, Tang Li, Zongyu Yin, Siyuan Feng, Ming Tu, Yuliang Ji,
Rui Xia, Mingbo Ma, Xuchen Song, Jitong Chen, Yuping Wang, Yuxuan Wang**
Speech, Audio & Music Intelligence (SAMI), ByteDance

## Abstract

Recent progress in music generation has been remarkably advanced by the state-of-the-art MusicLM, which comprises a hierarchy of three LMs, respectively, for semantic, coarse acoustic, and fine acoustic modelings. Yet, sampling with the MusicLM requires processing through these LMs one by one to obtain the fine-grained acoustic tokens, making it computationally expensive and prohibitive for a real-time generation. Efficient music generation with a quality on par with MusicLM remains a significant challenge. In this paper, we present **MeLoD**y (**M** for music; **L** for LM; **D** for diffusion), an LM-guided diffusion model that generates music audios of state-of-the-art quality meanwhile reducing 95.7% to 99.6% forward passes in MusicLM, respectively, for sampling 10s to 30s music. MeLoDy inherits the highest-level LM from MusicLM for semantic modeling, and applies a novel dual-path diffusion (DPD) model and an audio VAE-GAN to efficiently decode the conditioning semantic tokens into waveform. DPD is proposed to simultaneously model the coarse and fine acoustics by incorporating the semantic information into segments of latents effectively via cross-attention at each denoising step. Our experimental results suggest the superiority of MeLoDy, not only in its practical advantages on sampling speed and infinitely continuable generation, but also in its state-of-the-art musicality, audio quality, and text correlation.
Our samples are available at `https://Efficient-MeLoDy.github.io/`.

## 1  Introduction

Music is an art composed of harmony, melody, and rhythm that permeates every aspect of human life. With the blossoming of deep generative models [1–3], music generation has drawn much attention in recent years [4–6]. As a prominent class of generative models, language models (LMs) [7, 8] showed extraordinary modeling capability in modeling complex relationships across long-term contexts [9–11]. In light of this, AudioLM [3] and many follow-up works [5, 12–14] successfully applied LMs to audio synthesis. Concurrent to the LM-based approaches, diffusion probabilistic models (DPMs) [1, 15, 16], as another competitive class of generative models [2, 17], have also demonstrated exceptional abilities in synthesizing speech [18–20], sounds [21, 22] and music [6, 23].

However, generating music from free-form text remains challenging as the permissible music descriptions can be very diverse and relate to any of the genres, instruments, tempo, scenarios, or even some subjective feelings. Conventional text-to-music generation models are listed in Table 1, where both MusicLM [5] and Noise2Music [6] were trained on large-scale music datasets and demonstrated the state-of-the-art (SOTA) generative performances with high fidelity and adherence to various aspects of text prompts. Yet, the success of these two methods comes with large computational costs, which would be a serious impediment to their practicalities. In comparison, Moûsai [23] building upon DPMs made efficient samplings of high-quality music possible. Nevertheless, the number of their demonstrated cases was comparatively small and showed limited in-sample dynamics. Aiming for a feasible music creation tool, a high efficiency of the generative model is essential since it facilitates interactive creation with human feedback being taken into account as in [24].

37th Conference on Neural Information Processing Systems (NeurIPS 2023).

Table 1: A comparison of MeLoDy with conventional text-to-music generation models in the literature. We use **AC** to denote whether audio continuation is supported, **FR** to denote whether the sampling is faster than real-time on a V100 GPU, **VT** to denote whether the model has been tested and demonstrated using various types of text prompts including instruments, genres, and long-form rich descriptions, and **MP** to denote whether the evaluation was done by music producers.

| Model | Prompts | Training Data | AC | FR | VT | MP |
|---|---|---|---|---|---|---|
| Moûsai [23] | Text | 2.5k hours of music | ✓ | ✓ | ✗ | ✗ |
| MusicLM [5] | Text, Melody | 280k hours of music | ✓ | ✗ | ✓ | ✗ |
| Noise2Music [6] | Text | 340k hours of music | ✗ | ✗ | ✓ | ✗ |
| **MeLoDy** (Ours) | Text, Audio | 257k hours of music[1] | ✓ | ✓ | ✓ | ✓ |

While LMs and DPMs both showed promising results, we believe the relevant question is not whether one should be preferred over another but whether we can leverage both approaches with respect to their individual advantages, e.g., [25]. After analyzing the success of MusicLM, we leverage the highest-level LM in MusicLM, termed as *semantic LM*, to model the semantic structure of music, determining the overall arrangement of melody, rhythm, dynamics, timbre, and tempo. Conditional on this semantic LM, we exploit the non-autoregressive nature of DPMs to model the acoustics efficiently and effectively with the help of a successful sampling acceleration technique [26]. All in all, in this paper, we introduce several novelties that constitute our main contributions:

1. We present **MeLoDy** (**M** for music; **L** for LM; **D** for diffusion), an LM-guided diffusion model that generates music of competitive quality while reducing 95.7% and 99.6% iterations of MusicLM to sample 10s and 30s music, being faster than real-time on a V100 GPU.

2. We propose the novel dual-path diffusion (DPD) models to efficiently model coarse and fine acoustic information simultaneously with a particular semantic conditioning strategy.

3. We design an effective sampling scheme for DPD, which improves the generation quality over the previous sampling method in [23] proposed for this class of LDMs.

4. We reveal a successful audio VAE-GAN that effectively learns continuous latent representations, and is capable of synthesizing audios of competitive quality together with DPD.

## 2 Related Work

**Audio Generation**   Apart from the generation models shown in Table 1, there are also music generation models [28, 29] that can generate high-quality music samples at high speed, yet they cannot accept free-form text conditions and can only generate single-genre music, e.g., techno music in [29]. There also are some successful music generators in the industry, e.g. Mubert [30] and Riffusion [31], yet, as analyzed in [5], they struggled to compete with MusicLM in handling free-form text prompts. In a more general scope of audio synthesis, some promising text-to-audio synthesizers [12, 21, 22] trained with AudioSet [32] also demonstrated the ability to generate music from free-form text, but the musicality of their samples is limited.

**Acceleration of Autoregressive Models**   WaveNet [33] is a seminal work that demonstrates the capability of autoregressive (AR) models in generating high-fidelity audio. It comes with the drawback of extremely high computational cost in sampling. To improve its practical feasibility, Parallel WaveNet [34] and WaveRNN [35] were separately proposed to accelerate WaveNet. With a similar goal, our proposed MeLoDy can be viewed as an accelerated variant of MusicLM, where we replace the last two AR models with a dual-path diffusion model. Parallel to our work, SoundStorm [36] also exceedingly accelerates the AudioLM with a mask-based non-AR decoding scheme [37]. While it is applicable to MusicLM, the sound quality of this model is still limited by the bitrate of the neural codec. In comparison, the proposed diffusion model in MeLoDy operates with continuous-valued latent vectors, which by nature can be decoded into music audios of higher quality.

---

[1]We focus on non-vocal music data by using an audio classifier [27] to filter out in-house music data with vocals. Noticeably, generating vocals and instrumental music simultaneously in one model is defective even in the SOTA works [5, 6] because of the unnaturally sound vocals. While this work aims for generating production-level music, we improve the fidelity by reducing the tendency of generating vocals.

**Network Architecture** The architecture designed for our proposed DPD was inspired by the dual-path networks used in the context of audio separation, where Luo et al. [38] initiated the idea of segmentation-based dual-path processing, and triggered a number of follow-up works achieving the state-of-the-art results [39–43]. Noticing that the objective in diffusion models indeed can be viewed as a special case of source separation, this kind of dual-path architecture effectually provides us a basis for simultaneous coarse-and-fine acoustic modeling.

## 3    Background on Audio Language Modeling

This section provides the preliminaries that serve as the basis for our model. In particular, we briefly describe the audio language modeling framework and the tokenization methods used in MusicLM.

### 3.1    Audio Language Modeling with MusicLM

MusicLM [5] mainly follows the audio language modeling framework presented in AudioLM [3], where audio synthesis is viewed as a language modeling task over a hierarchy of coarse-to-fine audio tokens. In AudioLM, there are two kinds of tokenization for representing different scopes of audio:

- **Semantic Tokenization**: K-means over representations from SSL, e.g., w2v-BERT [44];
- **Acoustic Tokenization**: Neural audio codec, e.g., SoundStream [45].

To better handle the hierarchical structure of the acoustic tokens, AudioLM further separates the modeling of acoustic tokens into coarse and fine stages. In total, AudioLM defines three LM tasks: (1) semantic modeling, (2) coarse acoustic modeling, and (3) fine acoustic modeling.

We generally define the sequence of conditioning tokens as $\mathbf{c}_{1:T_{\mathrm{cnd}}} := [\mathbf{c}_1, \ldots, \mathbf{c}_{T_{\mathrm{cnd}}}]$ and the sequence of target tokens as $\mathbf{u}_{1:T_{\mathrm{tgt}}} := [\mathbf{u}_1, \ldots, \mathbf{u}_{T_{\mathrm{tgt}}}]$. In each modeling task, a Transformer-decoder language model parameterized by $\theta$ is tasked to solve the following autoregressive modeling problem:

$$p_\theta(\mathbf{u}_{1:T_{\mathrm{tgt}}}|\mathbf{c}_{1:T_{\mathrm{cnd}}}) = \prod_{j=1}^{T_{\mathrm{tgt}}} p_\theta(\mathbf{u}_j|[\mathbf{c}_1, \ldots, \mathbf{c}_{T_{\mathrm{cnd}}}, \mathbf{u}_1, \ldots, \mathbf{u}_{j-1}]), \tag{1}$$

where the conditioning tokens are concatenated to the target tokens as prefixes. In AudioLM, semantic modeling takes no condition; coarse acoustic modeling takes the semantic tokens as conditions; fine acoustic modeling takes the coarse acoustic tokens as conditions. The three corresponding LMs can be trained in parallel with the ground-truth tokens, but need to be sampled sequentially for inference.

### 3.1.1    Joint Tokenization of Music and Text with MuLan and RVQ

To maintain the merit of audio-only training, MusicLM relies on joint audio-text embedding model, termed as MuLan [46], which can be individually pre-trained with large-scale music data and weakly-associated, free-form text annotations. This MuLan model is learned to project the music audio and its corresponding text description into the same embedding space such that the paired audio-text embeddings can be as close as possible. In MusicLM, the embeddings of music and text are tokenized using a separately learned residual vector quantization (RVQ) [45] module. Then, to generate music from a text prompt, MusicLM takes the MuLan tokens from the RVQ as the conditioning tokens in the semantic modeling stage and the coarse acoustic modeling stage, following Eq. (1). Given the prefixing MuLan tokens, the semantic tokens, coarse acoustic tokens, and fine acoustic tokens can be subsequently computed by LMs to generate music audio adhering to the text prompt.

## 4    Model Description

The overall training and sampling pipelines of MeLoDy are shown in Figure 1, where, we have three modules for representation learning: (1) MuLan, (2) Wav2Vec2-Conformer, and (3) audio VAE, and two generative models: a language model (LM) and a dual-path diffusion (DPD) model, respectively, for semantic modeling and acoustic modeling. In the same spirit as MusicLM, we leverage LM to model the semantic structure of music for its promising capability of modeling complex relationships across long-term contexts [9–11]. We also similarly pre-train a MuLan model

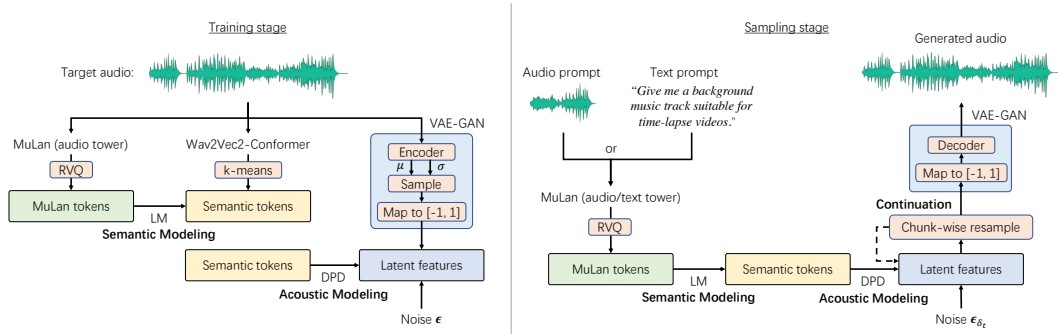

Figure 1: The training and sampling pipelines of MeLoDy

to obtain the conditioning tokens. For semantic tokenization, after empirically compared against w2v-BERT [44], we employ a Wav2Vec2-Conformer model, which follows the same architecture as Wav2Vec2 [47] but employs the Conformer blocks [48] in place of the Transformer blocks. The remainder of this section presents our newly proposed DPD model and the audio VAE-GAN used for DPD model, while other modules overlapped with MusicLM are described in Appendix B regarding their training and implementation details.

## 4.1 Audio VAE-GANs for Latent Representation Learning

To avoid learning arbitrarily high-variance latent representations, Rombach et al. [2] examined a KL-regularized image autoencoder for latent diffusion models (LDMs) and demonstrated extraordinary stability in generating high-quality image [49], igniting a series of follow-up works [50]. Such an autoencoder imposes a KL penalty on the encoder outputs in a way similar to VAEs [51, 52], but, different from the classical VAEs, it is adversarially trained as in generative adversarial networks (GANs) [53]. In this paper, this class of autoencoders is referred to as the *VAE-GAN*. Although VAE-GANs are promisingly applied in image generation, there are limited comparable attempts in audio generation. In this work, we propose to use a similarly trained VAE-GAN for raw audio.

Specifically, the audio VAE-GAN is trained to reconstruct 24kHz audio with a striding factor of 96, resulting in a 250Hz latent sequence. The architecture of the decoder is the same as that in HiFi-GAN [54]. For the encoder, we basically replace the up-sampling modules in the decoder with convolution-based down-sampling modules while other modules stay the same. For adversarial training, we use the multi-period discriminators in [54] and the multi-resolution spectrogram discriminators in [55]. The implementation and training details are further discussed in Appendix B.

## 4.2 Dual-Path Diffusion: Angle-Parameterized Continuous-Time Latent Diffusion Models

The proposed dual-path diffusion (DPD) model is a variant of diffusion probabilistic models (DPMs) [1, 15, 56] in continuous-time [16, 57–59]. Instead of directly operating on raw data space $\mathbf{x} \sim p_{\text{data}}(\mathbf{x})$, with reference to LDMs [2], DPD operates on a low-dimensional latent space $\mathbf{z}_0 = \mathcal{E}_\phi(\mathbf{x})$, such that the audio can be approximately reconstructed from the latent vectors: $\mathbf{x} \approx \mathcal{D}_\phi(\mathbf{z}_0)$, where $\mathcal{E}_\phi$ and $\mathcal{D}_\phi$ are the encoder and the decoder in VAE-GAN, respectively. Diffusing the latent space could significantly relieve the computational burden of DPMs [2]. Also, sharing a similar observation with [2], we find that audio VAE-GAN performed more stabler than other VQ-based autoencoders [45, 60] when working with the outputs from diffusion models.

Formally speaking, DPD is a Gaussian diffusion process $\mathbf{z}_t$ that is fully specified by two strictly positive scalar-valued, continuously differentiable functions $\alpha_t, \sigma_t$ [16]: $q(\mathbf{z}_t|\mathbf{z}_0) = \mathcal{N}(\mathbf{z}_t; \alpha_t \mathbf{z}_0, \sigma_t^2 \mathbf{I})$ for any $t \in [0, 1]$. In the light of [58], we define $\alpha_t := \cos(\pi t/2)$ and $\sigma_t := \sin(\pi t/2)$ to benefit from some nice trigonometric properties, i.e., $\sigma_t = \sqrt{1 - \alpha_t^2}$ (a.k.a. variance-preserving [16]). With this definition, the forward diffusion process of $\mathbf{z}_t$ can be re-parameterized in terms of angle $\delta \in [0, \pi/2]$:

$$\mathbf{z}_\delta = \cos(\delta)\mathbf{z}_0 + \sin(\delta)\boldsymbol{\epsilon}, \quad \boldsymbol{\epsilon} \sim \mathcal{N}(\mathbf{0}, \mathbf{I}), \tag{2}$$

which implies $\mathbf{z}_\delta$ gets noisier as the angle $\delta$ increases from 0 to $\pi/2$.

To create a generative process, a $\theta$-parameterized variational model $p_\theta(\mathbf{z}_{\delta-\omega}|\mathbf{z}_\delta)$ is trained to reverse the diffusion process by enabling taking any step $\omega \in (0, \delta]$ backward in angle. By discretizing $\pi/2$

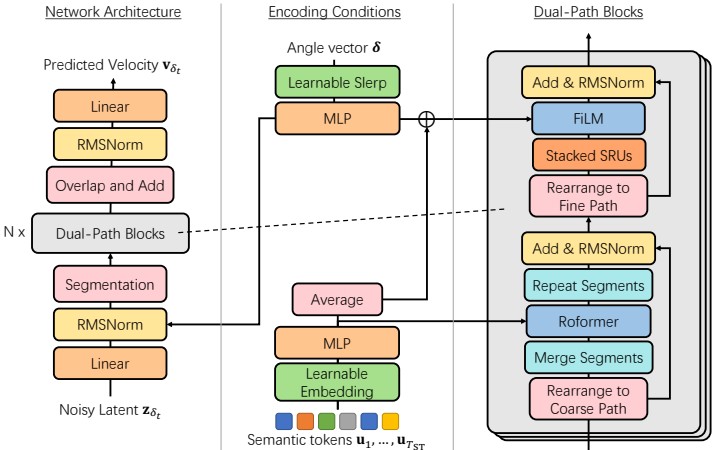

Figure 2: The proposed dual-path diffusion (DPD) model

into $T$ finite segments, we can generate $\mathbf{z}_0$ from $\mathbf{z}_{\pi/2} \sim \mathcal{N}(\mathbf{0}, \mathbf{I})$ in $T$ sampling steps:

$$p_\theta(\mathbf{z}_0|\mathbf{z}_{\pi/2}) = \int_{\mathbf{z}_{\delta_{1:T-1}}} \prod_{t=1}^{T} p_\theta(\mathbf{z}_{\delta_t - \omega_t}|\mathbf{z}_{\delta_t}) \, d\mathbf{z}_{\delta_{1:T-1}}, \quad \delta_t = \begin{cases} \frac{\pi}{2} - \sum_{i=t+1}^{T} \omega_i, & 1 \le t < T; \\ \frac{\pi}{2}, & t = T, \end{cases} \quad (3)$$

where $\omega_1, \dots, \omega_T$, termed as the *angle schedule*, satisfy $\sum_{t=1}^{T} \omega_t = \pi/2$. Regarding the choice of angle schedule, Schneider et al. [23] proposed a uniform one, i.e., $\omega_t = \frac{\pi}{2T}$ for all $t$. Yet, we observe that noise scheduling in the previous works [61, 62] tend to take larger steps at the beginning of the sampling followed smaller steps for refinement. With a similar perspective, we design another linear angle schedule, written as

$$\omega_t = \frac{\pi}{6T} + \frac{2\pi t}{3T(T+1)}, \quad (4)$$

which empirically gives more stable and higher-quality results. Appendix D presents the comparison results of this linear angle schedule against the uniform schedule used in [23].

### 4.2.1 Diffusion Velocity Prediction

In DPD, we model the diffusion velocity at $\delta$ [58], defined as $\mathbf{v}_\delta := \frac{d\mathbf{z}_\delta}{d\delta}$. It can be simplified as:

$$\mathbf{v}_\delta = \frac{d\cos(\delta)}{d\delta}\mathbf{z}_0 + \frac{d\sin(\delta)}{d\delta}\boldsymbol{\epsilon} = \cos(\delta)\boldsymbol{\epsilon} - \sin(\delta)\mathbf{z}_0. \quad (5)$$

When $\mathbf{v}_\delta$ is given, we can easily remedy the original sample $\mathbf{z}_0$ from a noisy latent $\mathbf{z}_\delta$ at any $\delta$, since $\mathbf{z}_0 = \cos(\delta)\mathbf{z}_\delta - \sin(\delta)\mathbf{v}_\delta$. This suggests $\mathbf{v}_\delta$ a feasible target for neural network prediction $\hat{\mathbf{v}}_\theta(\mathbf{z}_\delta; \mathbf{c})$, where $\mathbf{c}$ generally denotes the set of conditions controlling the music generation. In MeLoDy, as illustrated in Figure 1, the semantic tokens $\mathbf{u}_1, \dots, \mathbf{u}_{T_{ST}}$, which are obtained from the SSL model during training and generated by the LM at inference time, are used to condition the DPD model. In our experiments, we find that the stability of generation can be significantly improved if we use token-based discrete conditions to control the semantics of the music and let the diffusion model learn the embedding vector for each token itself. As in [23, 58], this velocity prediction network $\theta$ can be effectively trained with a mean squared error (MSE) loss:

$$\mathcal{L} := \mathbb{E}_{\mathbf{z}_0 \sim p_{\text{data}}(\mathbf{z}_0), \boldsymbol{\epsilon} \sim \mathcal{N}(\mathbf{0}, \mathbf{I}), \delta \sim \text{Uniform}[0,1]} \left[ \left\| \cos(\delta)\boldsymbol{\epsilon} - \sin(\delta)\mathbf{z}_0 - \hat{\mathbf{v}}_\theta(\cos(\delta)\mathbf{z}_0 + \sin(\delta)\boldsymbol{\epsilon}; \mathbf{c}) \right\|_2^2 \right], \quad (6)$$

which forms the basis of DPD's training loss.

### 4.2.2 Multi-Chunk Velocity Prediction

With reference to [23], for long-context generation, we can incrementally appending new chunks of random noise to infinitely continue audio generation. To achieve this, the velocity prediction

network needs to be trained to handle the chunked input, where each chunk exhibits a different scale of noisiness. In particular, we define the multi-chunk velocity target $\mathbf{v}_{\text{tgt}}$ that comprises $M$ chunks of velocities. Given $\mathbf{z}_0, \mathbf{z}_\delta, \boldsymbol{\epsilon} \in \mathbb{R}^{L \times D}$ with $L$ representing the length of latents and $D$ representing the latent dimensions, we have $\mathbf{v}_{\text{tgt}} := \mathbf{v}_1 \oplus \cdots \oplus \mathbf{v}_M$, where $\oplus$ is the concatenation operation and

$$\mathbf{v}_m := \cos(\delta_m)\boldsymbol{\epsilon}[L_{m-1} : L_m, :] - \sin(\delta_m)\mathbf{z}_0[L_{m-1} : L_m, :], \quad L_m := \left\lfloor \frac{mL}{M} \right\rfloor. \tag{7}$$

Here, we use the NumPy slicing syntax (0 as the first index) to locate the $m$-th chunk, and we draw $\delta_m \sim \text{Uniform}[0, \pi/2]$ for each chunk at each training step to determine the noise scale. The MSE loss in Eq. (6) is then extended to

$$\mathcal{L}_{\text{multi}} := \mathbb{E}_{\mathbf{z}_0, \boldsymbol{\epsilon}, \delta_1, \ldots, \delta_M} \left[ \| \mathbf{v}_{\text{tgt}} - \hat{\mathbf{v}}_\theta(\bar{\mathbf{z}}_{\delta_1} \oplus \cdots \oplus \bar{\mathbf{z}}_{\delta_M}; \mathbf{c}) \|_2^2 \right], \tag{8}$$

$$\bar{\mathbf{z}}_{\delta_m} := \cos(\delta_m)\mathbf{z}_0[L_{m-1} : L_m, :] + \sin(\delta_m)\boldsymbol{\epsilon}[L_{m-1} : L_m, :]. \tag{9}$$

Different from the original setting where we use a global noise scale for the network input [1, 61, 63], in the case of multi-chunk prediction, we need to specifically inform the network what the noise scales are for all $M$ chunks. Therefore, we append an angle vector $\boldsymbol{\delta}$ to the set of conditions $\mathbf{c} := \{\mathbf{u}_1, \ldots, \mathbf{u}_{T_{\text{ST}}}, \boldsymbol{\delta}\}$ to record the angles drawn in all $M$ chunks aligned with the $L$-length input:

$$\boldsymbol{\delta} := [\delta_1]_{r=1}^{L_1} \oplus \cdots \oplus [\delta_M]_{r=1}^{L_M} \in \mathbb{R}^L, \tag{10}$$

where $[a]_{r=1}^B$ denotes the operation of repeating a scalar $a$ for $B$ times to make a $B$-length vector.

### 4.2.3 Dual-Path Modeling for Efficient and Effective Velocity Prediction

To predict the multi-chunk velocity with $\hat{\mathbf{v}}_\theta$, we propose a dual-path modeling mechanism, which plays a prime role in DPD for efficient parallel processing along coarse and fine paths and effective semantic conditioning. Figure 2 presents the computation procedures of $\hat{\mathbf{v}}_\theta$, which comprises several critical modules that we present one by one below.

To begin with, we describe how the conditions $\{\mathbf{u}_1, \ldots, \mathbf{u}_{T_{\text{ST}}}, \boldsymbol{\delta}\}$ are processed in DPD:

**Encoding Angle Vector** First, we encode $\boldsymbol{\delta} \in \mathbb{R}^L$, which records the frame-level noise scales of latents. Instead of using the classical positional encoding [1], we use a spherical interpolation [64] between two learnable vectors $\mathbf{e}_{\text{start}}, \mathbf{e}_{\text{end}} \in \mathbb{R}^{256}$ using broadcast multiplications, denoted by $\otimes$:

$$\mathbf{E}_{\boldsymbol{\delta}} := \text{MLP}^{(1)} \left( \sin(\boldsymbol{\delta}) \otimes \mathbf{e}_{\text{start}} + \sin(\boldsymbol{\delta}) \otimes \mathbf{e}_{\text{end}} \right) \in \mathbb{R}^{L \times D_{\text{hid}}}, \tag{11}$$

where, for all $i$, $\text{MLP}^{(i)}(\mathbf{x}) := \text{RMSNorm}(\mathbf{W}_2^{(i)}\text{GELU}(\mathbf{x}\mathbf{W}_1^{(i)} + \mathbf{b}_1^{(i)}) + \mathbf{b}_2^{(i)})$ projects an arbitrary input $\mathbf{x} \in \mathbb{R}^{D_{\text{in}}}$ to $\mathbb{R}^{D_{\text{hid}}}$ using RMSNorm [65] and GELU activation [66] with learnable $\mathbf{W}_1^{(i)} \in \mathbb{R}^{D_{\text{in}} \times D_{\text{hid}}}$, $\mathbf{W}_2^{(i)} \in \mathbb{R}^{D_{\text{hid}} \times D_{\text{hid}}}$, $\mathbf{b}_1^{(i)}, \mathbf{b}_2^{(i)} \in \mathbb{R}^{D_{\text{hid}}}$, and $D_{\text{hid}}$ is hidden dimension.

**Encoding Semantic Tokens** The remaining conditions are the discrete tokens representing semantic information $\mathbf{u}_1, \ldots, \mathbf{u}_{T_{\text{ST}}}$. Following the typical approach for embedding natural languages [8], we directly use a lookup table of vectors to map any token $\mathbf{u}_t \in \{1, \ldots, V_{\text{ST}}\}$ into a real-valued vector $E(\mathbf{u}_t) \in \mathbb{R}^{D_{\text{hid}}}$, where $V_{\text{ST}}$ denotes the vocabulary size of the semantic tokens, i.e., the number of clusters in k-means for Wav2Vec2-Conformer. By stacking the vectors along the time axis and applying another MLP block, we obtain $\mathbf{E}_{\text{ST}} := \text{MLP}^{(2)} \left( [E(\mathbf{u}_1), \ldots, E(\mathbf{u}_{T_{\text{ST}}})] \right) \in \mathbb{R}^{T_{\text{ST}} \times D_{\text{hid}}}$.

Conditional on the computed embeddings $\mathbf{E}_{\boldsymbol{\delta}}$ and $\mathbf{E}_{\text{ST}}$, we next show how the network input, i.e., $\mathbf{z}_{\delta_t}$ for the case of having same noise scale $\delta_t$ for all chunks and $\bar{\mathbf{z}}_{\delta_1} \oplus \cdots \oplus \bar{\mathbf{z}}_{\delta_M}$ for the case of having different noise scales, is processed in DPD for velocity prediction. For the simplicity of notation, we use $\mathbf{z}_{\delta_t}$ to denote the network input here and below. $\mathbf{z}_{\delta_t}$ is first linearly transformed and added up with the angle embedding of the same shape: $\mathbf{H} := \text{RMSNorm} \left( \mathbf{z}_{\delta_t}\mathbf{W}_{\text{in}} + \mathbf{E}_{\boldsymbol{\delta}} \right)$, where $\mathbf{W}_{\text{in}} \in \mathbb{R}^{D \times D_{\text{hid}}}$ is learnable. Then, a crucial segmentation operation is applied for dual-path modeling.

**Segmentation** As illustrated in Figure 3, the segmentation module divides a 2-D input into $S$ half-overlapping segments each of length $K$, represented by a 3-D tensor $\mathbb{H} := [\mathbf{0}, \mathbf{H}_1, \ldots, \mathbf{H}_S, \mathbf{0}] \in \mathbb{R}^{S \times K \times D_{\text{hid}}}$, where $\mathbf{H}_s := \mathbf{H} \left[ \frac{(s-1)K}{2} : \frac{(s-1)K}{2} + K, : \right]$, and $\mathbb{H}$ is zero-padded such that we have

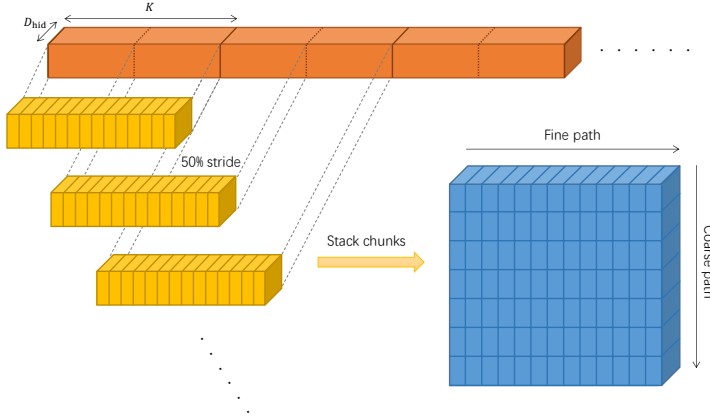

Figure 3: Diagram for visually understanding the segmentation operation for dual-path modeling

$S = \lceil \frac{2L}{K} \rceil + 1$. By choosing a segment size $K \approx \sqrt{L}$, the costs of sequence processing become sub-linear ($\mathcal{O}(\sqrt{L})$) as opposed to ($\mathcal{O}(L)$). This greatly reduces the difficulty of learning a very long sequence and permits MeLoDy to use higher-frequency latents for better audio quality. In this work, 250Hz latent sequences was used. In comparison, MusicLM [5] was built upon 50Hz codec.

**Dual-Path Blocks** After the segmentation, we obtain a 3-D tensor input. As shown in Figure 2, the tensor is subsequently passed to $N$ dual-path blocks, where each block contains two processing stages corresponding to coarse-path (i.e., inter-segment) and fine-path (i.e., intra-segment) processing, respectively. Similar to the observations in [40, 41], we find it superior to use an attention-based network for coarse-path processing and to use a bi-directional RNN for fine-path processing. The goal of fine acoustic modeling is to better reconstruct the fine details from the roughly determined audio structure [3]. At a finer scope, only the nearby elements matter and contain the most information needed for refinement, as supported by the modeling perspectives in neural vocoding [33, 35]. Specifically, we employ the Roformer network [67] for coarse-path processing, where we use a self-attention layer followed by a cross-attention layer to be conditional on $\mathbf{E}_{\text{ST}}$ with rotary positional embeddings. On the other hand, we use a stack of 2-layer simple recurrent units (SRUs) [68] for fine-path processing. A feature-wise linear modulation (FiLM) [69] layer is applied to the output of SRUs to assist the denoising with the angle embedding $\mathbf{E}_{\delta}$ and the pooled $\mathbf{E}_{\text{ST}}$. The details of inner mechanism in each dual-path block is presented in Appendix B.

### 4.2.4 Music Generation and Continuation

Suppose we have a well-trained multi-chunk velocity model $\hat{\mathbf{v}}_\theta$, we begin with a $L$-length latent generation, where $L$ is the length of latents we used in training. According to Appendix A, the DDIM sampling algorithm [26] can be re-formulated by applying the trigonometric identities:

$$\mathbf{z}_{\delta_t - \omega_t} = \cos(\omega_t)\mathbf{z}_{\delta_t} - \sin(\omega_t)\hat{\mathbf{v}}_\theta(\mathbf{z}_{\delta_t}; \mathbf{c}), \tag{12}$$

which, by running from $t = T$ to $t = 1$ using the defined $\omega_t$ in Eq. (4), we can generate a sample of $\mathbf{z}_0$ of length $L$. To continue generation, we append a new chunk composed of random noises to the generated $\mathbf{z}_0$ and drop the first chunk in $\mathbf{z}_0$. Recall that the inputs to $\hat{\mathbf{v}}_\theta$ are the $M$ concatenated noisy latents of different noise scales. The continuation of generation is feasible since the conditions (i.e., the semantic tokens and the angle vector) defined in DPD have an autoregressive nature at inference time. On one hand, the semantic tokens are generated by the semantic LM in an autoregressive manner, therefore we can continue the generation of semantic tokens for the new chunk. On the other hand, since the multi-chunk model $\hat{\mathbf{v}}_\theta$ is trained to tackle chunks of different noise scales with respect to the angle vector, we can simply ignore the generated audio (on the first $M - 1$ chunks) by zeroing the respective values and setting ones for the newly appended chunk, i.e., $\boldsymbol{\delta}_{\text{new}} := [0]_{r=1}^{L - \lceil L/M \rceil} \oplus [\delta_t]_{r=1}^{\lceil L/M \rceil}$. Then, the newly appended noise chunk can be transformed to meaningful music audio after $\lceil T/M \rceil$ step of DDIM sampling. For more details of generation, we present the corresponding algorithms in Appendix C. Besides music continuation, based on MuLan, MeLoDy also supports music prompts to generate music of a similar style, as shown in Figure 1. Examples of music continuation, and music prompting are shown on our demo page.

Table 2: The speed and the quality of our proposed MeLoDy on a CPU (Intel Xeon Platinum 8260 CPU @ 2.40GHz) or a GPU (NVIDIA Tesla V100) using different numbers of sampling steps.

| Steps ($T$) | Speed on CPU ($\uparrow$) | Speed on GPU ($\uparrow$) | FAD ($\downarrow$) | MCC ($\uparrow$) |
|---|---|---|---|---|
| (MusicCaps) | - | - | - | 0.43 |
| 5 | **1472Hz (0.06$\times$)** | **181.1kHz (7.5$\times$)** | 7.23 | 0.49 |
| 10 | 893Hz (0.04$\times$) | 104.8kHz (4.4$\times$) | 5.93 | 0.52 |
| 20 | 498Hz (0.02$\times$) | 56.9kHz (2.4$\times$) | **5.41** | **0.53** |

## 5 Experiments

### 5.1 Experimental Setup

**Data Preparation**  As shown in Table 1, MeLoDy was trained on 257k hours of music data (6.4M 24kHz audios), which were filtered with [27] to focus on non-vocal music. Additionally, inspired by the text augmentation in [6], we enriched the tag-based texts to generate music captions by asking ChatGPT [70]. This music description pool is used for the training of our 195.3M MuLan, where we randomly paired each audio with either the generated caption or its respective tags. In this way, we robustly improve the model's capability of handling free-form text.

**Semantic LM**  For semantic modeling, we trained a 429.5M LLaMA [71] with 24 layers, 8 heads, and 2048 hidden dimensions, which has a comparable number of parameters to that of the MusicLM [5]. For conditioning, we set up the MuLan RVQ using 12 1024-sized codebooks, resulting in 12 prefixing tokens. The training targets were 10s semantic tokens, which are obtained from discretizing the 25Hz embeddings from a 199.5M Wav2Vec2-Conformer with 1024-center k-means.

**Dual-Path Diffusion**  For the DPD model, we set the hidden dimension to $D_{hid} = 768$, and block number to $N = 8$, resulting in 296.6M parameters. For the input chunking strategy, we divide the 10s training inputs in a fixed length of $L = 2500$ into $M = 4$ parts. For segmentation, we used a segment size of $K = 64$ (i.e., each segment is 256ms long), leading to $S = 80$ segments. In addition, we applied the classifier-free guidance (CFG) [72] to DPD to improve the correspondence between samples and conditions. During training, the cross-attention to semantic tokens is randomly replaced by self-attention with a probability of 0.1. For sampling, the unconditional prediction $\mathbf{v}_{uncond}$ and the conditional prediction $\mathbf{v}_{cond}$ are linearly combined: $\rho\mathbf{v}_{cond} + (1 - \rho)\mathbf{v}_{uncond}$ with a scale of $\rho = 2.5$.

**Audio VAE-GAN**  For audio VAE-GAN, we used a hop size of 96, resulting in 250Hz latent sequences for encoding 24kHz music audio. The latent dimension $D = 16$, thus we have a total compression rate of $6\times$. The hidden channels used in the encoder were 256, whereas that used in the decoder were 768. The audio VAE-GAN in total contains 100.1M parameters.

### 5.2 Performance Analysis

**Objective Metrics**  We use the VGGish-based [73] Fréchet audio distance (FAD) [74] between the generated audios and the reference audios from MusicCaps [5] as a rough measure of generation fidelity.[2] To measure text correlation, we use the MuLan cycle consistency (MCC) [5], which calculates the cosine similarity between text and audio embeddings using a pre-trained MuLan.[3]

**Inference Speed**  We first evaluate the sampling efficiency of our proposed MeLoDy. As DPD permits using different numbers of sampling steps depending on our needs, we report its generation speed in Table 2. Surprisingly, MeLoDy steadily achieved a higher MCC score than that of the reference set, even taking only 5 sampling steps. This means that (i) the MuLan model determined that our generated samples were more correlated to MusicCaps captions than reference audios, and (ii) the proposed DPD is capable of consistently completing the MuLan cycle at significantly lower costs than the nested LMs in [5].

---

[2]Note that MeLoDy was mainly trained with non-vocal music data, its sample distribution could not fit the reference one as well as in [5, 6], since about 76% audios in MusicCaps contain either vocals or speech.

[3]Since our MuLan model was trained with a different dataset, our MCC results cannot be compared to [5, 6].

Table 3: The comparison of MeLoDy with the SOTA text-to-music generation models. **NFE** is the number of function evaluations [58] for generating $T$-second audio.[5] **Musicality**, **Quality**, and **Text Corr.** are the winning proportions in terms of musicality, quality, and text correlation, respectively.

| Model | NFE ($\downarrow$) | Musicality ($\uparrow$) | | Quality ($\uparrow$) | | Text Corr. ($\uparrow$) | |
|---|---|---|---|---|---|---|---|
| | | MLM | N2M | MLM | N2M | MLM | N2M |
| MusicLM [5] | $(25 + 200 + 400)T$ | **0.541** | - | 0.465 | - | **0.548** | - |
| Noise2Music [6] | $1000 + 800 + 800$ | - | **0.555** | - | 0.436 | - | **0.572** |
| **MeLoDy** (20 steps) | $25T + 20$ | 0.459 | 0.445 | **0.535** | **0.564** | 0.452 | 0.428 |

**Comparisons with SOTA models**   We evaluate the performance of MeLoDy by comparing it to MusicLM [5] and Noise2Music [6], which both were trained large-scale music datasets and demonstrated SOTA results for a wide range of text prompts. To conduct fair comparisons, we used the same text prompts in their demos (70 samples from MusicLM; 41 samples from Noise2Music),[4] and asked seven music producers to select the best out of a pair of samples or voting for a tie (both win) in terms of musicality, audio quality, and text correlation. In total, we conducted 777 comparisons and collected 1,554 ratings. We detail the evaluation protocol in Appendix F. Table 3 shows the comparison results, where each category of ratings is separated into two columns, representing the comparison against MusicLM (MLM) or Noise2Music (N2M), respectively. Finally, MeLoDy consistently achieved comparable performances (all winning proportions fall into [0.4, 0.6]) in musicality and text correlation to MusicLM and Noise2Music. Regarding audio quality, MeLoDy outperformed MusicLM ($p < 0.05$) and Noise2Music ($p < 0.01$), where the $p$-values were calculated using the Wilcoxon signed-rank test. We note that, to sample 10s and 30s music, MeLoDy only takes 4.32% and 0.41% NFEs of MusicLM, and 10.4% and 29.6% NFEs of Noise2Music, respectively.

**Diversity Analysis**   Diffusion models are distinguished for its high diversity [25]. We conduct an additional experiment to study the diversity and validity of MeLoDy's generation given the same text prompt of open description, e.g., feelings or scenarios. The sampled results were shown on our demo page, in which we obtained samples with diverse combinations of instruments and textures.

**Ablation Studies**   We also study the ablation on two aspects of the proposed method. In Appendix D, we compared the uniform angle schedule in [23] and the linear one proposed in DPD using the MCC metric and case-by-case qualitative analysis. It turns out that our proposed schedule tends to induce fewer acoustic issues when taking a small number of sampling steps. In Appendix E, we showed that the proposed dual-path architecture outperformed other architectures [23, 31] used for LDMs in terms of the signal-to-noise ratio (SNR) improvements using a subset of the training data.

## 6   Discussion

**Limitation**   We acknowledge the limitations of our proposed MeLoDy. To prevent from having any disruption caused by unnaturally sound vocals, our training data was prepared to mostly contain non-vocal music only, which may limit the range of effective prompts for MeLoDy. Besides, the training corpus we used was unbalanced and slightly biased towards pop and classical music. Lastly, as we trained the LM and DPD on 10s segments, the dynamics of a long generation may be limited.

**Broader Impact**   We believe our work has a huge potential to grow into a music creation tool for music producers, content creators, or even normal users to seamlessly express their creative pursuits with a low entry barrier. MeLoDy also facilitates an interactive creation process, as in Midjourney [24], to take human feedback into account. For a more precise tune of MeLoDy on a musical style, the LoRA technique [75] can be potentially applied to MeLoDy, as in Stable Diffusion [49].

---

[4]All samples for evaluation are available at https://Efficient-MeLoDy.github.io/. Note that our samples were not cherry-picked, whereas the samples we compared were cherry-picked [6], constituting very strong baselines.

[5]We use + to separate the counts for the iterative modules, i.e., LM or DPM. Suppose the cost of each module is comparable, then the time steps taken by LM and the diffusion steps taken by DPM can be fairly compared.

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
