# A Mathematical Background for Dual-Path Diffusion

## A.1 Forward Diffusion Process

In dual-path diffusion (DPD), we consider a Gaussian diffusion process [16] that continuously diffuses our generation target $\mathbf{z}_0$ into increasingly noisy versions of $\mathbf{z}_0$, denoted as $\mathbf{z}_t$ with $t \in [0, 1]$ running from $t = 0$ (least noisy) to $t = 1$ (most noisy). This forward diffusion process is formally defined as

$$q(\mathbf{z}_t|\mathbf{z}_0) = \mathcal{N}(\mathbf{z}_t; \alpha_t \mathbf{z}_0, \sigma_t^2 \mathbf{I}), \tag{13}$$

where two strictly positive scalar-valued, continuously differentiable functions $\alpha_t, \sigma_t$ define the noise schedule [1] of this forward diffusion process. Building upon the nice properties of Gaussian distributions, we can express $q(\mathbf{z}_t|\mathbf{z}_s)$, for any $0 \leq s < t \leq 1$, as another Gaussian distribution:

$$q(\mathbf{z}_t|\mathbf{z}_s) = \mathcal{N}\left(\mathbf{z}_t; \frac{\alpha_t}{\alpha_s}\mathbf{z}_s, \left(\sigma_t^2 - \frac{\alpha_t}{\alpha_s}\sigma_s^2\right)\mathbf{I}\right). \tag{14}$$

Regarding the choice of noise scheduling functions, we consider the typical setting used in [1, 15]: $\alpha_t = \sqrt{1 - \sigma_t^2}$, which gives rise to a *variance-preserving* diffusion process [16]. Specifically, we employ the trigonometric functions in [58], defined as follows:

$$\alpha_t := \cos(\pi t/2) \quad \sigma_t := \sin(\pi t/2) \quad \forall t \in [0, 1] \tag{15}$$

$$\Leftrightarrow \alpha_\delta := \cos(\delta) \quad \sigma_\delta := \sin(\delta) \quad \forall \delta \in [0, \pi/2]. \tag{16}$$

With this re-parameterization, the diffusion process can now be defined in terms of angle $\delta \in [0, \pi/2]$:

$$\mathbf{z}_\delta = \cos(\delta)\mathbf{z}_0 + \sin(\delta)\boldsymbol{\epsilon}, \quad \boldsymbol{\epsilon} \sim \mathcal{N}(\mathbf{0}, \mathbf{I}), \tag{17}$$

where $\mathbf{z}_\delta$ gets noisier as $\delta$ increases from 0 to $\pi/2$.

## A.2 Prediction of Diffusion Velocity

The diffusion velocity of $\mathbf{z}_\delta$ at $\delta$ [58] is defined as:

$$\mathbf{v}_\delta := \frac{d\mathbf{z}_\delta}{d\delta} = \frac{d\cos(\delta)}{d\delta}\mathbf{z}_0 + \frac{d\sin(\delta)}{d\delta}\boldsymbol{\epsilon} = \cos(\delta)\boldsymbol{\epsilon} - \sin(\delta)\mathbf{z}_0. \tag{18}$$

Based on $\mathbf{v}_\delta$, we can compute $\mathbf{z}_0$ and $\boldsymbol{\epsilon}$ from a noisy latent $\mathbf{z}_\delta$:

$$\mathbf{z}_0 = \cos(\delta)\mathbf{z}_\delta - \sin(\delta)\mathbf{v}_\delta = \alpha_\delta \mathbf{z}_\delta - \sigma_\delta \mathbf{v}_\delta; \tag{19}$$

$$\boldsymbol{\epsilon} = \sin(\delta)\mathbf{z}_\delta + \cos(\delta)\mathbf{v}_\delta = \sigma_\delta \mathbf{z}_\delta + \alpha_\delta \mathbf{v}_\delta, \tag{20}$$

which suggests $\mathbf{v}_\delta$ a feasible target for network prediction $\hat{\mathbf{v}}_\theta(\mathbf{z}_\delta; \mathbf{c})$ given a collection of conditions $\mathbf{c}$, as an alternative to the $\mathbf{z}_0$ prediction ($\hat{\mathbf{z}}_\theta(\mathbf{z}_\delta; \mathbf{c})$), e.g., in [16], and the $\boldsymbol{\epsilon}$ prediction ($\hat{\boldsymbol{\epsilon}}_\theta(\mathbf{z}_\delta; \mathbf{c})$), e.g., in [1, 61, 63]. As reported by Salimans and Ho [58] and Schneider et al. [23], training the neural network $\theta$ with a mean squared error (MSE) loss as in the pioneering work [1] remains effective:

$$\mathcal{L} := \mathbb{E}_{\mathbf{z}_0 \sim p_{\text{data}}(\mathbf{z}_0), \boldsymbol{\epsilon} \sim \mathcal{N}(\mathbf{0}, \mathbf{I}), \delta \sim \text{Uniform}[0, 1]} \left[ \|\cos(\delta)\boldsymbol{\epsilon} - \sin(\delta)\mathbf{z}_0 - \hat{\mathbf{v}}_\theta(\cos(\delta)\mathbf{z}_0 + \sin(\delta)\boldsymbol{\epsilon}; \mathbf{c})\|_2^2 \right], \tag{21}$$

which forms the basis of DPD's training loss, i.e., the simplest case of considering only a single chunk per input ($M = 1$) in Eq. (9). We can easily extend this to a multi-chunk version by sampling $M$ different angles $\delta_1, \ldots, \delta_M \sim \text{Uniform}[0, 1]$, where the $m$-th sampled angle is applied to the corresponding chunk of the latent, i.e., $\mathbf{z}_0[(m-1)L/M : mL/M]$.

## A.3 Generative Diffusion Process

Generation is done by inverting the forward process from a noise vector randomly drawn from $\mathcal{N}(\mathbf{0}, \mathbf{I})$. One efficient way to accomplish this is to take advantage of DDIM [26], which enables running backward from angle $\delta$ to angle $\delta - \omega$, for any step size $0 < \omega < \delta$:

$$p_\theta(\mathbf{z}_{\delta-\omega}|\mathbf{z}_\delta) := q\left(\mathbf{z}_{\delta-\omega}\Big|\mathbf{z}_0 = \frac{\mathbf{z}_\delta - \sigma_\delta \hat{\boldsymbol{\epsilon}}_\theta(\mathbf{z}_\delta; \mathbf{c})}{\alpha_\delta}\right) = \alpha_{\delta-\omega}\left(\frac{\mathbf{z}_\delta - \sigma_\delta \hat{\boldsymbol{\epsilon}}_\theta(\mathbf{z}_\delta; \mathbf{c})}{\alpha_\delta}\right) + \sigma_{\delta-\omega}\boldsymbol{\epsilon}, \tag{22}$$

where $\epsilon \sim \mathcal{N}(\mathbf{0}, \mathbf{I})$. Song et al. [26] considered $\epsilon \equiv \hat{\epsilon}_\theta(\mathbf{z}_\delta; \mathbf{c})$, leading to a deterministic update rule:

$$\mathbf{z}_{\delta-\omega} = \frac{\alpha_{\delta-\omega}}{\alpha_\delta}\mathbf{z}_\delta + \left(\sigma_{\delta-\omega} - \frac{\alpha_{\delta-\omega}\sigma_\delta}{\alpha_\delta}\right)\hat{\epsilon}_\theta(\mathbf{z}_\delta; \mathbf{c}). \tag{23}$$

By changing the prediction target to diffusion velocity, Salimans and Ho [58] reformulated DDIM as

$$p_\theta(\mathbf{z}_{\delta-\omega}|\mathbf{z}_\delta) := q\left(\mathbf{z}_{\delta-\omega} | \mathbf{z}_0 = \alpha_\delta\mathbf{z}_\delta - \sigma_\delta\hat{\mathbf{v}}_\theta(\mathbf{z}_\delta; \mathbf{c})\right) \tag{24}$$

$$= \alpha_{\delta-\omega}\left(\alpha_\delta\mathbf{z}_\delta - \sigma_\delta\hat{\mathbf{v}}_\theta(\mathbf{z}_\delta; \mathbf{c})\right) + \sigma_{\delta-\omega}\epsilon, \tag{25}$$

where $\epsilon \sim \mathcal{N}(\mathbf{0}, \mathbf{I})$. Here, we can similarly consider a parameterized noise vector $\epsilon \equiv \sigma_\delta\mathbf{z}_\delta + \alpha_\delta\hat{\mathbf{v}}_\theta(\mathbf{z}_\delta; \mathbf{c})$ based on Eq. (20), yielding a simplified deterministic update rule:

$$\mathbf{z}_{\delta-\omega} = \alpha_{\delta-\omega}\left(\alpha_\delta\mathbf{z}_\delta - \sigma_\delta\hat{\mathbf{v}}_\theta(\mathbf{z}_\delta; \mathbf{c})\right) + \sigma_{\delta-\omega}\left(\sigma_\delta\mathbf{z}_\delta + \alpha_\delta\hat{\mathbf{v}}_\theta(\mathbf{z}_\delta; \mathbf{c})\right) \tag{26}$$

$$= \left(\alpha_{\delta-\omega}\alpha_\delta - \sigma_{\delta-\omega}\sigma_\delta\right)\mathbf{z}_\delta + \left(\sigma_{\delta-\omega}\alpha_\delta - \alpha_{\delta-\omega}\sigma_\delta\right)\hat{\mathbf{v}}_\theta(\mathbf{z}_\delta; \mathbf{c}) \tag{27}$$

$$= \cos(\omega)\mathbf{z}_\delta - \sin(\omega)\hat{\mathbf{v}}_\theta(\mathbf{z}_\delta; \mathbf{c}) \tag{28}$$

where the last equation is obtained by applying the trigonometric identities:

$$\alpha_{\delta-\omega}\alpha_\delta - \sigma_{\delta-\omega}\sigma_\delta = \cos(\delta-\omega)\cos(\delta) - \sin(\delta-\omega)\sin(\delta) = \cos(\omega); \tag{29}$$

$$\sigma_{\delta-\omega}\alpha_\delta - \alpha_{\delta-\omega}\sigma_\delta = \sin(\delta-\omega)\cos(\delta) - \cos(\delta-\omega)\sin(\delta) = \sin(\omega). \tag{30}$$

Building upon this angular update rule and having specified the angle step sizes $\omega_1, \ldots, \omega_T$ with $\sum_{t=1}^{T}\omega_t = \pi/2$, we can generate samples from $\mathbf{z}_{\pi/2} \sim \mathcal{N}(\mathbf{0}, \mathbf{I})$ after $T$ steps of sampling:

$$\mathbf{z}_{\delta_t - \omega_t} = \cos(\omega_t)\mathbf{z}_{\delta_t} - \sin(\omega_t)\hat{\mathbf{v}}_\theta(\mathbf{z}_{\delta_t}; \mathbf{c}), \quad \delta_t = \begin{cases} \frac{\pi}{2} - \sum_{i=t+1}^{T}\omega_i, & 1 \le t < T; \\ \frac{\pi}{2}, & t = T, \end{cases} \tag{31}$$

running from $t = T$ to $t = 1$.

## B   Training and Implementation Details

### B.1   Wav2Vec2-Conformer

Our implementation of Wav2Vec2-Conformer was based on an open-source library.[6] In particular, Wav2Vec2-Conformer follows the same architecture as Wav2Vec2 [47], but replaces the Transformer structure with the Conformer [48]. This model with 199.5M parameters was trained in self-supervised learning (SSL) manner similar to [47] using our prepared 257k hours of music data.

### B.2   MuLan

Our reproduced MuLan [46] is composed of a music encoder and a text encoder. For music encoding, we rely on a publicly accessible Audio Spectrogram Transformer (AST) model pre-trained on AudioSet,[7] which gives promising results on various audio classification benchmarks. For text encoding, we employ the BERT [8] base model pre-trained on a large corpus of English data using a masked language modeling (MLM) objective.[8] These two pre-trained encoders, together having 195.3M parameters, were subsequently fine-tuned on the 257k hours of music data with a text augmentation technique similar to [6]. In particular, we enriched the tag-based texts to generate music captions by asking ChatGPT [70]. At training time, we randomly paired each audio with either the generated caption or its respective tags. In practice, this could robustly improve the model's capability of handling free-form text.

### B.3   Audio VAE-GAN

As shown in Figure 4, we train a VAE-GAN to extract 250Hz 16-dimensional latent $\mathbf{z} \in \mathbb{R}^{L \times 16}$ from a 24kHz audio $\mathbf{x} \in \mathbb{R}^{T_{\text{wav}}}$ with $L = \lceil T_{\text{wav}}/96 \rceil$. The audio VAE-GAN mainly comprises three

---

[6]https://huggingface.co/docs/transformers/model_doc/wav2vec2-conformer
[7]https://huggingface.co/MIT/ast-finetuned-audioset-10-10-0.4593
[8]https://huggingface.co/bert-base-uncased

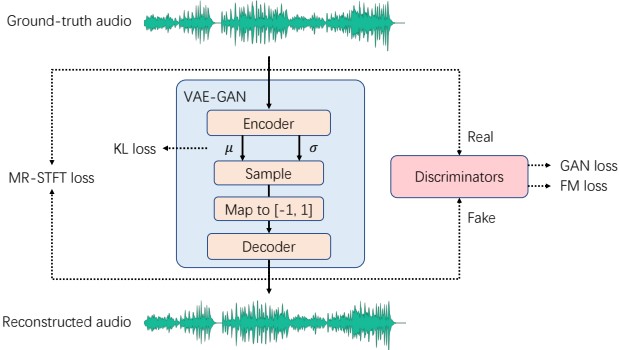

Figure 4: The audio VAE-GAN trained for dual-path diffusion models

trainable modules: (i) a variational Gaussian encoder $\mathcal{E}_\phi(\mathbf{x}) \equiv \mathcal{N}(\mu_\phi(\mathbf{x}), \sigma_\phi(\mathbf{x})\mathbf{I})$, (ii) a decoder $\mathcal{D}_\phi(\mathbf{z})$, and (iii) a set of $n$ discriminators $\{D^{(i)}\}_{i=1}^n$.

Regarding network architecture, we use the ResNet-style convolutional neural networks (CNNs) in HiFi-GAN [54] as the backbone.[9] For the encoder, we replace the up-sampling blocks in HiFi-GAN with convolution-based down-sampling blocks, with down-sampling rates of $[2, 3, 4, 4]$, output dimensions of $[32, 64, 128, 256]$ and kernel sizes of $[5, 7, 9, 9]$ in four down-sampling blocks, giving 40M parameters. The final layer of the encoder maps the 256-dimensional output to two 16-dimensional latent sequences, respectively for the mean and variance of diagonal Gaussian sampling.[10] As shown in Figure 4, to match the normal range of targets for diffusion models [1, 2], we map the sampled outputs to $[-1, 1]$ by $\mathbf{z}_{(i,j)} := \min\left\{\max\left\{\bar{\mathbf{z}}_{(i,j)}/3, -1\right\}, 1\right\} \forall i, j$, where the subscript $(i, j)$ denotes the value on the $i$-th row and $j$-th column, and the choice of 3 in practice would sieve extreme values occupying $< 0.1\%$. For the architecture setting of the decoder, it inherits the same architecture of HiFi-GAN, and uses up-sampling rates of $[4, 4, 3, 2]$, kernel sizes of $[9, 9, 5, 7]$ and larger number of output channels ($[768, 384, 192, 96]$) for four up-sampling blocks, taking 60.1M parameters.

For adversarial training, we use the multi-period discriminators in [54] and the multi-resolution spectrogram discriminators in [55]. The training scheme is similar to that in [54]. The training loss for the encoder and the decoder comprises four components:

$$\mathcal{L}_{\text{vae-gan}}(\phi) := \mathbb{E}_{\mathbf{x} \sim p_{\text{data}}(\mathbf{x})} \left[ \mathbb{E}_{\mathbf{z} \sim \mathcal{E}_\phi(\mathbf{x})} \left[ \lambda_{\text{mr-stft}} \mathcal{L}_{\text{mr-stft}} + \lambda_{\text{fm}} \mathcal{L}_{\text{fm}} + \lambda_{\text{gan}} \mathcal{L}_{\text{gan}} + \lambda_{\text{kl}} \mathcal{L}_{\text{kl}} \right] \right] \tag{32}$$

$$\mathcal{L}_{\text{mr-stft}} := \sum_{r=1}^{R} \left\| \text{STFT}_r(\mathbf{x}) - \text{STFT}_r(\mathcal{D}_\phi(\mathbf{z})) \right\|_1 \tag{33}$$

$$\mathcal{L}_{\text{fm}} := \sum_{i=1}^{n} \frac{1}{|D^{(i)}|} \sum_{l=1}^{|D^{(i)}|} \left\| D_l^{(i)}(\mathbf{x}) - D_l^{(i)}(\mathcal{D}_\phi(\mathbf{z})) \right\|_1 \tag{34}$$

$$\mathcal{L}_{\text{gan}} := \sum_{i=1}^{n} \left( D^{(i)}(\mathcal{D}_\phi(\mathbf{z})) - 1 \right)^2 \tag{35}$$

$$\mathcal{L}_{\text{kl}} := \text{KL}\left( \mathcal{E}_\phi(\mathbf{x}) || \mathcal{N}(\mathbf{0}, \mathbf{I}) \right), \tag{36}$$

where $\text{STFT}_r$ computes the magnitudes after the $r$-th short-time Fourier transform (STFT) out of $R = 7$ STFTs (the number of FFTs $= [8192, 4096, 2048, 512, 128, 64, 32]$; the window sizes $= [4096, 2048, 1024, 256, 64, 32, 16]$; the hop sizes $= [2048, 1024, 512, 128, 32, 16, 8]$), $|D^{(i)}|$ denotes the number of hidden layers used for feature matching in discriminator $D^{(i)}$, $D_l^{(i)}$ denotes the outputs of the $l$-th hidden layers in discriminator $D^{(i)}$, and $\lambda_{\text{mr-stft}}, \lambda_{\text{fm}}, \lambda_{\text{gan}}, \lambda_{\text{kl}}$ are the weights, respectively, for the multi-resolution STFT loss $\mathcal{L}_{\text{mr-stft}}$, the feature matching loss $\mathcal{L}_{\text{fm}}$, the GAN's generator loss $\mathcal{L}_{\text{gan}}$, and the Kullback–Leibler divergence based regularization loss $\mathcal{L}_{\text{kl}}$. To balance the scale of different losses, we set $\lambda_{\text{mr-stft}} = 50$, $\lambda_{\text{fm}} = 20$, $\mathcal{L}_{\text{gan}} = 1$, and $\lambda_{\text{kl}} = 5 \times 10^{-3}$ in our training. In

---

[9]Our implementation is similar to that in https://github.com/jik876/hifi-gan.

[10]The Gaussian sampling is referred to LDMs' implementation at https://github.com/CompVis/latent-diffusion/blob/main/ldm/modules/distributions/distributions.py

practice, we find it critical to lower the scale of the KL loss for a better reconstruction, though the distribution of the latents can still be close to zero mean and unit variance.

## B.4 Dual-Path Blocks

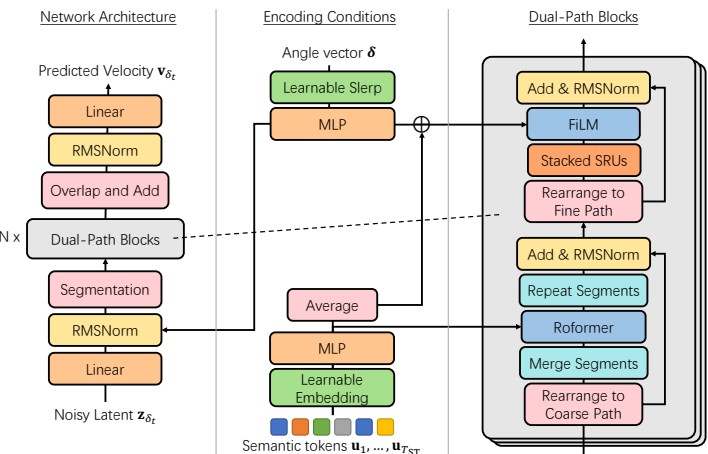

Figure 5: The proposed dual-path diffusion (DPD) model

Recall that, after the segmentation, we obtain $S$ half-overlapping segments each of length $K$, represented by a 3-D tensor $\mathbb{H} := [\mathbf{0}, \mathbf{H}_1, \ldots, \mathbf{H}_S, \mathbf{0}] \in \mathbb{R}^{S \times K \times D_{\text{hid}}}$, where $\mathbf{H}_s :=$ $\mathbf{H}\left[\frac{(s-1)K}{2} : \frac{(s-1)K}{2} + K, :\right]$, and $\mathbb{H}$ is zero-padded such that we have $S = \left\lceil \frac{2L}{K} \right\rceil + 1$. As shown in Figure 5, the tensor is subsequ for $N$ dual-path blocks, each block exhibits an architecture shown on the rightmost of . The input to the $i$-th dual-path block is denoted as $\mathbb{H}^{(i)}$, and we have $\mathbb{H}^{(1)} := \mathbb{H}$. Each block contains two stages corresponding to coarse-path (i.e., inter-segment) and fine-path (i.e., intra-segment) processing, respectively. Similar to the observations in [40, 41], we find it superior to use an attention-based network for coarse-path processing and to use a bi-directional RNN for fine-path processing. The goal of fine acoustic modeling is to better reconstruct the fine details from the roughly determined audio structure [3]. At a finer scope, only the nearby elements matter and contain the most information needed for refinement, as supported by the modeling perspectives in neural vocoding [33, 35]. Specifically, we employ the Roformer network [67] for coarse-path processing, where we use a self-attention layer followed by a cross-attention layer to be conditional on $\mathbf{E}_{\text{ST}}$ with rotary positional embedding. On the other hand, we use a stack of 2-layer simple recurrent units (SRUs) [68] for fine-path processing. The feature-wise linear modulation (FiLM) [69] is applied to the output of SRUs to assist the denoising with the angle embedding $\mathbf{E}_\delta$ and the pooled $\mathbf{E}_{\text{ST}}$. Each of these processing stages is detailed below.

**Coarse-Path Processing** In a dual-path block, we first process the coarse path corresponding to the vertical axis shown in Figure 3, in which the columns are processed in parallel:

$$\mathbb{H}_{\text{c-out}}^{(i)} := \text{RepeatSegments}\left(\left[\text{Roformer}\left(\text{MergeSegments}\left(\mathbb{H}^{(i)}\right)[:, k, :]\right), k = 0, \ldots, K_{\text{MS}}^{(i)} - 1\right]\right),$$
(37)

where the coarse-path output $\mathbb{H}_{\text{c-out}}^{(i)} \in \mathbb{R}^{S \times K \times D_{\text{hid}}}$ has the same shape as $\mathbb{H}^{(i)}$, and $\text{MergeSegments}(\cdot)$ and $\text{RepeatSegments}(\cdot)$ are the operations that, respectively, compress and expand the segments horizontally to aggregate the information within a segment for a coarser scale of inter-segment processing. Note that, without taking the merging and repeating operations, the vertical axis is simply a sequence formed by skipping $K/2$ elements in $\mathbf{H}$, which does not really capture the desired coarse information. The merging is done by averaging every pair of $2^{\min\{i, N-i+1\}}$ columns with zero paddings and a half stride such that $K_{\text{MS}}^{(i)} = \left\lceil \frac{K}{2^{\min\{i, N-i+1\}-1}} \right\rceil$. The upper part of Figure 6 illustrates the case of $i = 2$. Similar to [41], our definition of $K_{\text{MS}}^{(i)}$ changes the width of the 3d tensor with the block index $i$ in a sandglass style, as we have the shortest segment at the middle block and the longest segment at the first and the last block. To match with the original length, a repeating operation following from the Roformer is performed, as shown in the lower part of Figure 6.

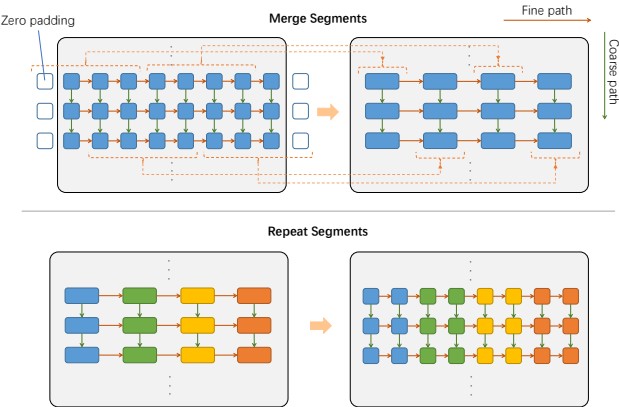

Figure 6: Merge Segments and Repeat Segments

**Fine-Path Processing** We then obtain the fine-path input: $\mathbb{H}^{(i)}_{\text{f-in}} := \text{RMSNorm}\left(\mathbb{H}^{(i)} + \mathbb{H}^{(i)}_{\text{c-out}}\right)$, which is fed to a two-layer SRU by parallelly processing the rows illustrated in Figure 3:

$$\mathbb{H}^{(i)}_{\text{f-out}} := \left[\text{FiLM}\left(\text{SRU}\left(\mathbb{H}^{(i)}_{\text{f-in}}[s,:,:]\right), \mathbf{E}_{\boldsymbol{\delta}}\left[\frac{sL}{S},:\right] + \frac{1}{T_{\text{ST}}}\sum_{t=0}^{T_{\text{ST}}-1}\mathbf{E}_{\text{ST}}[t,:]\right), s = 0, \dots, S-1\right], \tag{38}$$

where $\text{FiLM}(\mathbf{x}, \mathbf{m}) := \text{MLP}_3\left((\mathbf{x} \otimes \text{MLP}_1(\mathbf{m})) + \text{MLP}_2(\mathbf{m})\right)$ for an arbitrary input $\mathbf{x}$ and modulation condition $\mathbf{m}$, and $\otimes$ is the operations of broadcast multiplication. Followed from this, we have the input for the next dual-path block: $\mathbb{H}^{(i+1)} := \text{RMSNorm}\left(\mathbb{H}^{(i)}_{\text{f-in}} + \mathbb{H}^{(i)}_{\text{f-out}}\right)$. After recursively processing through $N$ dual-path blocks, the 3-D tensor is transformed back to a 2-D matrix using an overlap-and-add method [38]. Finally, the predicted velocity is obtained as follows:

$$\hat{\mathbf{v}}_\theta(\mathbf{z}_{\delta_t}; \mathbf{c}) := \text{RMSNorm}\left(\text{OverlapAdd}\left(\mathbb{H}^{(N+1)}\right)\right)\mathbf{W}_{\text{out}}, \tag{39}$$

where $\mathbf{W}_{\text{out}} \in \mathbb{R}^{D_{\text{hid}} \times D}$ is learnable.

## C  Algorithms for MeLoDy

---

**Algorithm 1** Music Generation

1: **given** $\mathcal{D}_\phi, \hat{\mathbf{v}}_\theta, T, \omega_1, \dots, \omega_T$
2: **input** Music/text prompt $\mathcal{P}$
3:
4: Initialize $\delta_T = \pi/2$
5: Compute the MuLan tokens for $\mathcal{P}$: $\mathbf{c}_{1:T_{\text{cnd}}}$
6: Generate $\mathbf{u}_{1:T_{\text{ST}}}$ from $\mathbf{c}_{1:T_{\text{cnd}}}$ with LM ▷ (1)
7: Sample $\mathbf{z}_{\delta_T} \sim \mathcal{N}(\mathbf{0}, \mathbf{I})$
8: **for** $t = T$ to 1 **do**
9:  Prepare condition: $\mathbf{c} = \{\mathbf{u}_{1:T_{\text{ST}}}, [\delta_t]_{r=1}^L\}$ ▷ (10)
10:  Update angle: $\delta_{t-1} = \delta_t - \omega_t$ ▷ (3)
11:  $\mathbf{z}_{\delta_{t-1}} = \cos(\omega_t)\mathbf{z}_{\delta_t} - \sin(\omega_t)\hat{\mathbf{v}}_\theta(\mathbf{z}_{\delta_t}; \mathbf{c})$ ▷ (12)
12: **end for**
13: **repeat**
14:  **pass** $\mathbf{c}_{1:T_{\text{cnd}}}$, $\mathbf{u}_{1:T_{\text{ST}}}$ and $\mathbf{z}_0$ to **Algorithm 2**
15: **until** $\mathbf{z}_0$ reaches the desired length
16: **return** $\mathcal{D}_\phi(\mathbf{z}_0)$

---

**Algorithm 2** Music Continuation

1: **given** $\mathcal{D}_\phi, \hat{\mathbf{v}}_\theta, T, M, \omega_1, \dots, \omega_T$
2: **input** Music $\mathbf{z}_0$ and $\mathbf{c}_{1:T_{\text{cnd}}}$, $\mathbf{u}_{1:T_{\text{ST}}}$ (if provided)
3:
4: Denote $M_{\text{ST}} = \lceil T_{\text{ST}}/M \rceil$, $L_M = \lceil L/M \rceil$
5: Initialize $\delta_T = \pi/2$
6: Generate $\mathbf{u}_{T_{\text{ST}}:T_{\text{ST}}+M_{\text{ST}}}$ from $\mathbf{c}_{1:T_{\text{cnd}}} \oplus \mathbf{u}_{M_{\text{ST}}:T_{\text{ST}}}$
7: Sample $\mathbf{z}_{\text{new}} \sim \mathcal{N}(\mathbf{0}, \mathbf{I}) \in \mathbb{R}^{L_M}$
8: Save first chunk: $\mathbf{z}_{\text{save}} = \mathbf{z}_0[: L_M]$
9: $\mathbf{z}_{\delta_T} = \mathbf{z}_0[L_M :] \oplus \mathbf{z}_{\text{new}}$
10: **for** $t = T$ to 1 **do**
11:  Update $\boldsymbol{\delta}_{\text{new}} = [0]_{r=1}^{L-L_M} \oplus [\delta_t]_{r=1}^{L_M}$
12:  Prepare condition: $\mathbf{c} = \{\mathbf{u}_{M_{\text{ST}}:T_{\text{ST}}+M_{\text{ST}}}, \boldsymbol{\delta}_{\text{new}}\}$
13:  Update angle: $\delta_{t-1} = \delta_t - \omega_t$
14:  $\mathbf{z}_{\delta_{t-1}} = \cos(\omega_t)\mathbf{z}_{\delta_t} - \sin(\omega_t)\hat{\mathbf{v}}_\theta(\mathbf{z}_{\delta_t}; \mathbf{c})$
15: **end for**
16: **return** $\mathbf{z}_{\text{save}} \oplus \mathbf{z}_0$

---

MeLoDy supports music or text prompting for music generation, as illustrated in Figure 1. We concretely detail the sampling procedures in Algorithm 1, where the algorithm starts by generating

the latent sequence of length $L$ and then recursively prolongs the latent sequence using Algorithm 2 until it reaches the desired length.

We further explain how music continuation can be effectively done in DPD. Recall that the inputs for training DPD are the concatenated chunks of noisy latents in different noise scales. To continue a given music audio, we can add a new chunk composed of random noises and drop the first chunk. This is feasible since the conditions (i.e., the semantic tokens and the angles) defined for DPD have an autoregressive nature. Based on the semantic LM, we can continue the generation of $\lceil T_{\text{ST}}/M \rceil$ semantic tokens for the new chunk. Besides, it is sensible to keep the chunks other than the new chunk to have zero angles: $\boldsymbol{\delta}_{\text{new}} := [0]_{r=1}^{L-\lceil L/M \rceil} \oplus [\delta_t]_{r=1}^{\lceil L/M \rceil}$, as shown in Algorithm 2.

In addition, music inpainting can be done in a similar way. We replace the inpainting partition of the input audio with random noise and partially set the angle vector to zeros to mark the positions where the denoising operations are not needed. Yet, in this case, the semantic tokens can only be roughly estimated using the remaining part of the music audio.

# D    Ablation Study on Angle Schedules

Table 4: The objective measures for the ablation study on angle schedules.

| Angle schedule ($\omega_t$) | Steps ($T$) | FAD ($\downarrow$) | MCC ($\uparrow$) |
|---|---|---|---|
| Uniform [23]: $\omega_t = \frac{\pi}{2T}$ | 10 | 8.52 | 0.45 |
| | 20 | 6.31 | 0.49 |
| Ours proposed in Eq. (4): $\omega_t = \frac{\pi}{6T} + \frac{2\pi t}{3T(T+1)}$ | 10 | **5.93** | **0.52** |
| | 20 | **5.41** | **0.53** |

We conduct an ablation study on angle schedules to validate the effectiveness of our proposed angle schedule $\omega_1, \ldots, \omega_T$ in Eq. (4) in comparison to the previous uniform angle schedule [23] also used for angle-parameterized continuous-time diffusion models. In particular, the same pre-trained DPD model $\hat{\mathbf{v}}_\theta$ and was used to sample with two different angle schedules with 10 steps and 20 steps, respectively, conditional on the same semantic tokens generated for the text prompts in MusicCaps. Table 4 shows their corresponding objective measures in terms of FAD and MCC. We observe a significant improvement, especially when taking a small number of sampling steps, by using the proposed sampling method. This is aligned with our expectations that taking larger steps at the beginning of the sampling followed by smaller steps could improve the quality of samples, similar to the findings in previous diffusion scheduling methods [61, 62].

We further qualitatively analyze the quality of the generated samples using some simple text prompts of instruments, i.e., flute, saxophone, and acoustic guitar, by pair-wise comparing their spectrograms as illustrated in Figure 7. In the case of "flute", sampling with the proposed angle schedule results in a piece of naturally sound music, being more saturated in high-frequency bands and even remedying the breathiness of flute playing, as shown in Figure 7b. On the contrary, we can observe from the spectrogram in Figure 7a that the sample generated with a uniform angle schedule is comparatively monotonous. In the case of "saxophone", the uniform angle schedule leads to metallic sounds that are dissonant, as revealed by the higher energy in 3kHz to 6kHz frequency bands shown in Figure 7c. In comparison, the frequency bands are more consistent in Figure 7d, when the proposed schedule is used. While the comparatively poorer sample quality using the uniform schedule could be caused by the limited number of sampling steps, we also show the spectrograms after increasing the sampling steps from 10 to 20. In the case of "acoustic guitar", when taking 20 sampling steps, the samples generated with both angle schedules sound more natural. However, in Figure 7e, we witness a horizontal line around the 4.4kHz frequency band, which is unpleasant to hear. Whereas, the sample generated by our proposed schedule escaped such an acoustic issue, as presented in Figure 7f.

# E    Ablation Study on Architectures

To examine the superiority of our proposed dual-path model in Figure 2, we also study the ablation of network architectures. In particular, to focus on the denoising capability of different architectures,

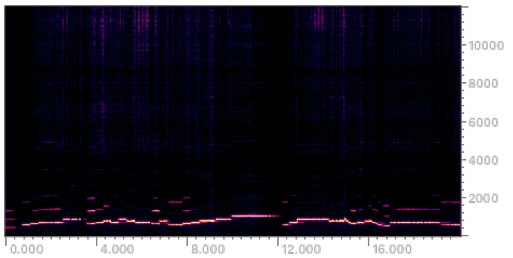 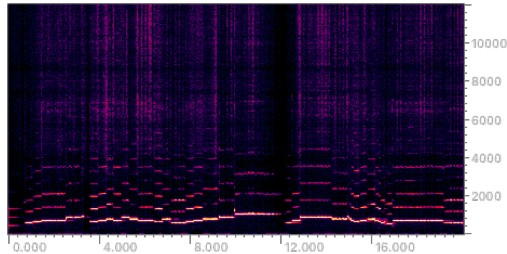

(a) 10-step sampling with uniform angle schedule for text prompt: "flute"

(b) 10-step sampling with our proposed angle schedule for text prompt: "flute"

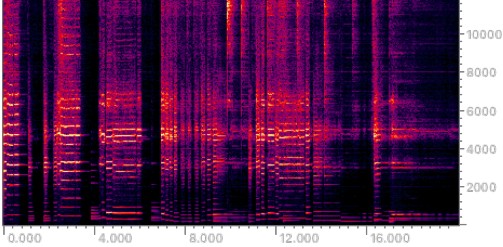 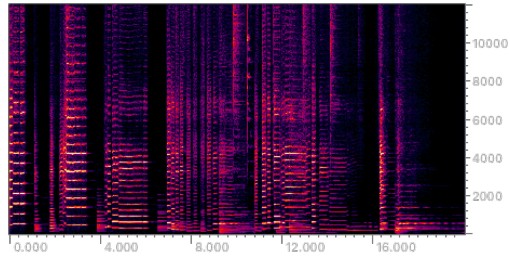

(c) 10-step sampling with uniform angle schedule for text prompt: "saxophone"

(d) 10-step sampling with our proposed angle schedule for text prompt: "saxophone"

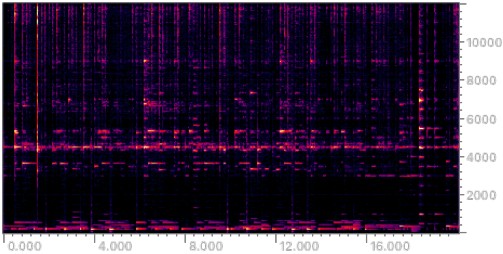 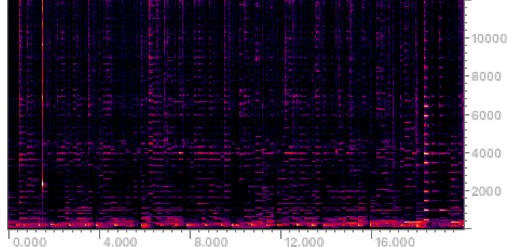

(e) 10-step sampling with uniform angle schedule for text prompt: "acoustic guitar"

(f) 10-step sampling with our proposed angle schedule for text prompt: "acoustic guitar"

Figure 7: Spectrograms of generated samples with uniform (left) and our proposed (right) angle schedules

Table 5: The objective measures for the ablation study on architectures.

| Architecture | Velocity MSE ($\downarrow$) | SI-SNRi ($\uparrow$) |
|---|---|---|
| UNet-1D [23] | 0.13 | 5.33 |
| UNet-2D [31] | 0.15 | 4.96 |
| **DPD** (Ours) | **0.12** | **6.15** |

we only take a subset of the training data (approximately 5k hours of music data) to train different networks with the same optimization configurations – 100k training steps using AdamW optimizer with a learning rate of $5 \times 10^{-4}$ and a batch size of 96 on 8 NVIDIA V100 GPUs. For a fair comparison, we train the UNet-1D[11] and the UNet-2D[12] with comparable numbers of parameters (approximately 300M). Note that the FAD and MCC measures are not suitable for evaluating the performance of each forward pass of the trained network for denoising. In addition to the training objective, i.e., the velocity MSE, we use the scale-invariant signal-to-noise ratio (SNR) improvements (SI-SNRi) [38, 40] between the true latent $\mathbf{z}$ and the predicted latent $\hat{\mathbf{z}} = \alpha_\delta \mathbf{z}_\delta - \sigma_\delta \hat{\mathbf{v}}_\theta(\mathbf{z}_\delta; \mathbf{c})$. The results are shown in Table 5, where our proposed dual-path architecture outperforms the other two widely used UNet-style architectures in terms of both the velocity MSE and SI-SNRi.

---

[11]Our implementation of UNet-1D relied on https://github.com/archinetai/a-unet.

[12]Our implementation of UNet-2D relied on https://huggingface.co/riffusion/riffusion-model-v1.

# F Qualitative Evaluation

To conduct a pair-wise comparison, each music producer is asked to fill in the form composed of three questions. Specifically, we present the user interface for each pair-wise comparison in Figure 8.

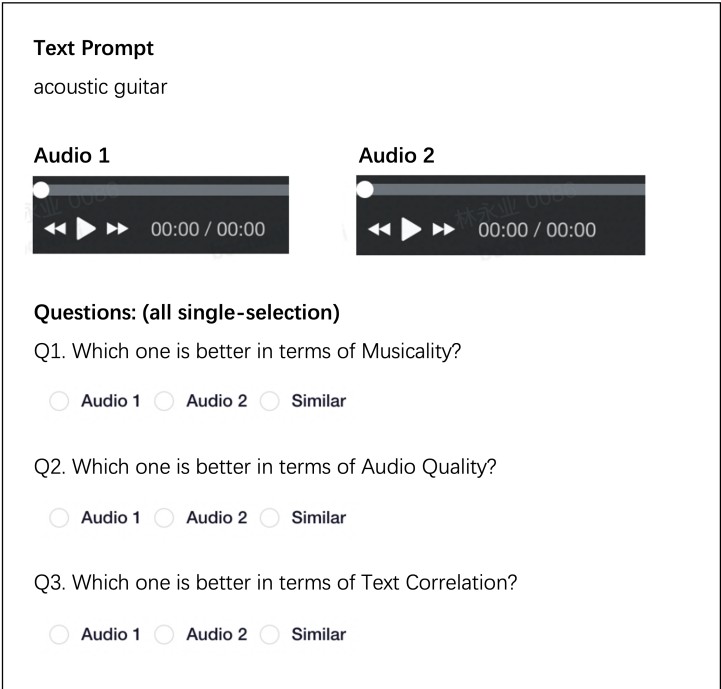

Figure 8: The user interface for music producers in each pair-wise comparison