# OpenReview forum: "Efficient Neural Music Generation"
_NeurIPS.cc/2023/Conference — NeurIPS 2023 poster_

### Official Review · Reviewer_KL1t · 2023-07-05

**Soundness:** 2 fair
**Presentation:** 2 fair
**Contribution:** 3 good
**Rating:** 4
**Confidence:** 4

**Summary:**

This paper introduces an LM-guided diffusion model designed for the generation of music audios. The proposed system follows a three-step process, starting with the extraction of audio-text embeddings (MuLan tokens) using a pre-trained MuLan model. Next, semantic tokens (wav2vec tokens) are generated with the aid of a language model. Finally, the synthesized music audio is produced using a dual-path diffusion model.

Experimental results highlight the superiority of the proposed model, showcasing its practical advantages in terms of sampling speed and the ability to generate music seamlessly.

**Strengths:**

1. The paper exhibits clear and well-written sections before the experiments, ensuring the reader’s comprehension and engagement.
2. The paper addresses a highly meaningful topic and task, aligning with the current trends and hot topics in language modeling.
3. The proposed method in this paper is well-designed, effectively integrating domain knowledge in music and demonstrating a deep understanding of the subject matter.

**Weaknesses:**

1. The overall structure of the paper’s writing could benefit from optimization. For example, important experimental results should ideally be included in the main body rather than being relegated to the appendix.
2. The paper lacks a comprehensive objective evaluation of the main experiments, particularly in comparing the proposed method with the baseline approaches instead of only using objective metrics in the ablation study.
3. Based on the findings in Table 3, the proposed method exhibits a noticeable drop in musicality and text correlation compared to the baseline approaches. This raises questions about the motivation behind the proposed method. Is the trade-off in performance for the sake of efficiency justified? Does the efficiency gained by the proposed method outweigh the potential loss in important aspects such as musicality and text correlation?
4. The appendix file is difficult to locate, as there is no indication or hint for readers that it is available on the demo page. Clear guidance or references within the paper would greatly assist reviewers and readers in accessing the relevant appendix materials.
5. The paper lacks a clear explanation for certain design choices in the proposed method, such as the motivation behind utilizing an angle parameterization for the diffusion schedule and the specific design of Equation 4. Without a detailed rationale and justification for these design decisions, it becomes difficult to comprehend their significance and how they improve the overall performance and effectiveness of the method.

**Questions:**

NA

---

> ### Author Rebuttal · Authors · 2023-08-08
>
> We are grateful to the the reviewer's comments and suggestions. In the following, we address each of the concerns raised by the reviewer.
>
> ### Response to concern #1
>
> **Optimizing overall structure of this paper**: We are thankful for the reviewer's constructive comment, which is taken seriously to revise this paper for a clearer presentation. Thanks to the concrete suggestions raised by all reviewers, we have already made several changes in the revised paper. Some of the amendments are described below:
>
> - (1) We have re-organized Section 4 to make it more condense and self-contained. A lot of details regarding network architecture has been moved to the Appendix. We put more efforts to summarize the idea in sentences instead of equations. Also, some leaps of definition were avoided in the revised paper. For example, the diffusion velocity was ill-defined in Section 4 (as pointed out by reviewer **4HjZ** and **ieEr**). We added a brief description in advance to make it self-contained.
>
> - (2) As we make the Section 4 to be more condense, the ablation studies is now moved from the Appendix to the main paper, as suggested the reviewer.
>
> - (3) We also extended the related work to add more discussion on audio/music generation methods as well as audio representation learning, as suggested the reviewer **cbjn**.
>
> ---
>
> ### Response to concern #2
>
> **Lack of comprehensive objective evaluation in the main experiments**: For each of the individual modules, we were eager to investigate suitable objective measures for leave-one-out rigorous comparisons, e.g., SI-SNR for different diffusion architectures. However, for an overall assessment on the whole generation system, since the competing systems are not publicly released and were trained with entirely different datasets, the fairness of objective comparison cannot be guaranteed.
>
> Besides, as insisted in the title of this paper, MeLoDy is proposed to solve the computational problem associated with the SOTA MusicLM, and to make efficient music generation from free-form text possible. We focus on the comparison against MusicLM in terms of various aspects -- speed, quality, musicality and text correlation. Among these 4 aspects, only the assessment of speed was objectively evaluated since, to the best of our knowledge, no known objective measures can be as precise as the subjective evaluations of music professionals when assessing the audio quality, musicality and text correlation of an arbitrary audio sample.
>
> ---
>
> ### Response to concern #3
>
> **``noticeable drop in musicality and text correlation''**: First, we would like to note that the baseline samples we used for comparisons are (at least partially) cherrypicked, as stated in the main paper of Noise2Music. In contrast, we used the non-cherrypicked ones for all the text prompts taken from MusicLM and Noise2Music. In fact, the number of samples released on our demo page is more than 300, which prohibits the chance of cherrypicking sample for each prompt. Presumably, the sample variances of diffusion and LM could cover marginal differences in musicality and text correlation. In other words, from a practical sense, if we need to improve the musicality and text correlation given a text prompt, we can re-sample with the same prompt until we get a desired sample. Whereas, the generation speed and audio quality (denoting upper bound of the generation quality) aspects are rather steady -- they would not drastically vary even if we draw for more samples.
>
> ---
>
> ### Response to concern #4
>
> **Appendix**: We would like to apologize for the missing appendix due to some sudden network issues happened around the appendix deadline. We chose to immediately upload it to the demo page (the commit time in Github is tractable). Although the supplementary material is regarded as optional, the reviewer is right that we strongly recommend the readers to look into the Appendix for a thorough understanding of this paper. We will definitely supplement the Appendix file if this paper can be accepted.
>
> ---
>
> ### Response to concern #5
>
> **Lack of clear explanation for certain design choices**: We acknowledge the reviewer's comment regarding presentation clarity. A similar argument has been made by reviewer **cbjn**. We take this seriously, and add more texts for explaining the motivations and the reasons behind the choice of each module (e.g. angle parameterization has been compared against noise parameterization and performed the best in [1]). As pointed out before, we have restructured Section 4 to provide more incentives in textual forms instead of mathematical forms. We believe our revisions made for this paper sufficiently improve the clarity and address the problems concerned by the reviewer.
>
> [1] Flavio Schneider, et al. Moûsai: Text-to-music generation with long-context latent diffusion. 2023.

---

### Official Review · Reviewer_4HjZ · 2023-07-05

**Soundness:** 3 good
**Presentation:** 2 fair
**Contribution:** 3 good
**Rating:** 6
**Confidence:** 5

**Summary:**

The authors present a text to music model, with the aim of achieveing faster than real time generation on a V100 gpu. The authors introduce a latent space diffusion model. The latent space is obtained with an adverdarial VAE (replacing the VQ VAE from MusicLM), and a first LM is trained on the semantic tokens just as in MusicLM. The LM is also conditioned on the quantized output of a Mulan model. Then a latent diffusion model is trained to predict the latent representation of the VAE, conditioned on the semantic tokens. In order to improve the speed of the model, the authors uses a dual path architecture, i.e. they alternate between modeling long range and short range dependencies. Short range dependencies are modeled with a simple RNN, while long range ones are modeled with an attention based model.
The model is trained on ~260k hours of music with tag labels, enriched with ChatGPT outputs into full descriptions.

The authors compare with the publicly available samples of MusicLM and Noise2Music. They show that their model achieves better quality but overall worse musicality and text fidelity. They also report the FAD on MusicCaps (5.1) which is worse than that of MusicLM (4.0) or Noise2Music (2.1).

**Strengths:**

- large improvement in terms of speed compared with the many models and diffusion steps required by Noise2Music.
- improvements on the diffusion schedule.
- interesting idea of using a dual path architecture to improve the runtime. This allows to take full advantage of the non causal aspect of the diffusion process used here instead of auto-regressive modeling.
- Runtimes on Table 2 are impressive.
- subjective evaluations show improved quality.


**Weaknesses:**

- still requires a large number of models to train, e.g. Mulan model, a semantic model w2v-bert + one LM for the semantic tokens, and one dual path diffusion model, without counting the VAE.
- ablation studies are limited to the noise schedule. Not strong objective motivation for most of the design choices.
- authors mention that the Mulan Cycle consistency score is worse for the test set MusicCaps than their model. However that could be an artefact of how the authors' Mulan model is trained, e.g. on weak labels generated from partial tags rather than full descriptions, in particular considering that their own model is trained with this Mulan model, so that it could overfit any bias it has.
- when comparing the number of function calls in Table 3, the authors forget to mention that for MusicLM, each call require the evaluation of a single time step, while their own model require to process all the timesteps of the input sequence. In particular, in the context of batched generation, this can make a big difference between the two models.
- objective metric FAD is worse than existing methods, and two of the subjective metrics (musicality and text correlation). Subjective evaluations are done on the samples released by the other methods which might be better than the average though.

Note that the authors provide their supplementary material through a website, which might not be an acceptable way of doing so, as it doesn't enforce the deadline for the supplementary material submission.



**Questions:**

What are "velocities" ? this term is not properly defined when introduced.

Why are the authors using different noise levels for different segments? I see in the supplementary this is used for inpainting. Other than that regular sampling would always use the same noise level for all segments? Is there a benefit to this training method for regular sampling?

In Section 5.1, the authors mention they filter the data to focus on non vocal music. Is the 257k hours number given before or after this filtering?

The sample rate of the proposed of the VAE seems relatively large compared to the one used in MusicLM, is this something the authors experimented with?

**Limitations:**

The authors fail to discuss potential adverse effect of automation of creative jobs. The authors fail to discuss the potential breaches to intellectual property for the data used. In particular, the authors should discuss how the dataset was assembled.

---

> ### Author Rebuttal · Authors · 2023-08-08
>
> We appreciate the reviewer's careful reading of our paper. We hope the following fully address all concerns mentioned by the reviewer:
>
> ### Response to concern #1
>
> **``large number of models to train''**: The reviewer is correct that MeLoDy comprises **5 components** for a successful training, i.e., a semantic LM, a diffusion model (DPD), a SSL module (Wav2Vec2-Conformer), a prompt encoder (MuLan) and an autoencoder (VAE-GAN). Yet, in comparison to the SOTA MusicLM, the design MeLoDy is already simpler, since MusicLM consists of training **6 components**: 3 LMs (semantic, coarse & fine), a SSL module (w2v-BERT), a prompt encoder (MuLan) and an autoencoder (SoundStream). In this sense, we could reckon this work as one step towards a less complex system.
>
> ---
>
> ### Response to concern #2
>
> **Motivations behind the design choices**: Noticeably, reviewer **cbjn** and **KL1t** also share a similar argument. We take these comments seriously, and restructure Section 4 with several amendments:
> - (1) We re-organized Section 4 to make it more condense and self-contained. We added textual explanations for the design choice of each module (e.g. angle parameterization has been compared against noise parameterization and performed the best in [1]). A lot of mathematical details regarding network architecture has been moved to the Appendix.
> - (2) Some leaps of definition were avoided in the revised paper. For example, the diffusion velocity was ill-defined in Section 4, as pointed out by the reviewer.
> - (3) As we compress Section 4 to be more condense, the ablation studies is now moved from the Appendix to the main paper, as suggested the reviewer **KL1t**.
>
> Besides, the reviewer mentioned the **objective motivation** for the design choices. In fact, for each of the individual modules, we investigated suitable objective measures for rigorous comparisons, e.g., SI-SNR for diffusion architectures. However, for other modules like SSL, it is very hard to tell which one is better with an objective measure (if there is any, please let us know.) We therefore can only listen to some random samples (10-100 per system) led by different SSL modules (i.e., Wav2Vec2, Wav2Vec2-Conformer, and w2v-BERT), and subjectively vote for a better module. In common practice, this has been a simple yet effective way to select an audio generative model, since no objective metric can compete with human ears in audio quality assessment.
>
> [1] Flavio Schneider, et al. Moûsai: Text-to-music generation with long-context latent diffusion. 2023.
>
> ---
>
> ### Response to concern #3
>
> **Mulan Cycle Consistency**: The reviewer is right that there is an inductive bias in the MCC measurement. Yet, we note that the objective of using MCC here and in MusicLM was to examine whether the conditioning MuLan embeddings form a cycle in the conditional generation (MuLan -> MeLoDy -> MuLan). The MCC results in Table 2 was meant to confirm that our proposed DPD is capable of consistently completing the MuLan cycle at significantly lower costs.
>
> ---
>
> ### Response to concern #4
>
> **Fairness of comparing the number of function calls**: The reviewer's argument is thought-provoking. Yet, we note that each function call in an LM also need to take previous tokens into account. In the case of decoder-only LM, the computational cost of each call is $\mathcal{O}(L^2)$ with $L$ being the length of tokens. It is actually comparable to the cost of each call of a diffusion model ($\mathcal{O}(L^2)$ if we also use attention in diffusion). Let alone the big-O notation, in practice the cost of each call of a DPD is in fact much lower than that of a Transformer-based LM, since the segmentation leads to $L=L'/K\approx \sqrt{L'}$, where $L'$ is the original sequence length. The cost of attention in DPD thus becomes almost linear to the length of sequence.
>
> ---
>
> ### Response to concern #5
>
> **``objective metric FAD is worse than existing methods''**: Notably, it is unfair to directly compare the FAD of MeLoDy against MusicLM and Noise2Music, since MeLoDy was mainly trained on processed non-vocal data and most samples in MusicCaps contain speech or vocal.
>
> ---
>
> ### Response to concern #6
>
> **Worse subjective metrics (musicality and text correlation)**: First, we would like to note that the baseline samples we used for comparisons are (at least partially) cherrypicked, as stated in the main paper of Noise2Music. In contrast, we used the non-cherrypicked ones for all the text prompts taken from MusicLM and Noise2Music. In fact, the number of samples released on our demo page is more than 300, which prohibits the chance of cherrypicking sample for each prompt. Presumably, the sample variances of diffusion and LM could cover marginal differences in musicality and text correlation. In other words, from a practical sense, if we need to improve the musicality and text correlation given a text prompt, we can re-sample with the same prompt until we get a desired sample. Whereas, the generation speed and audio quality (denoting upper bound of the generation quality) aspects are rather steady -- they would not drastically vary even if we draw for more samples.
>
> ---
>
> ### Response to other questions
>
> - The definition of velocities ($\mathbf{v}\_t:=\frac{\partial \mathbf{z}\_t}{\partial t}$) has been added to the main context.
> - Using different noise levels for different segments makes possible the infinitely continuable generation (see the detailed response to reviewer **hBX7**).
> - The 257k hours of data has already gone through the filtering.
> - Yes, we have tried a number of different configurations for VAE-GAN, and we can conclude that output rate of the encoder (a.k.a. frequency of the latent sequence) is especially important, as the audio reconstruction from a higher-frequency latent sequence (e.g. 250Hz in MeLoDy) appears to be much better than a lower-frequency one (e.g. 50Hz in MusicLM). We have added this justification in our revised paper.

---

> > ### Comment · Reviewer_4HjZ · 2023-08-19
> > **about the complexity**
> >
> > Regarding concern #4: this is only true if the attention is dominating the run time, which should be experimentally validated.
> >
> > I don't see any novel elements regarding ablation studies, although the authors state they have reorganised the paper. Having no new elements, I will not increase my score.
> >
> > Regarding the multiple noise levels, another possibility is to use the same noise level everywhere but to do "teacher forcing" on the already generated segments, i.e. discarding the output of the model for those segment, and using the previously obtained output, interpolated with the proper amount of noise. It would be interesting to compare the two, as the "teacher forcing" approach has a simpler training procedure.
> >
> > Regarding the VAE-GAN: if the authors have any results on that, it would make the paper stronger to include those.

---

> > > ### Author Response · Authors · 2023-08-20
> > > **Response to the reviewer**
> > >
> > > Thanks for the reviewer's insightful comments.
> > >
> > > We agree to the reviewer's suggestion on providing additional empirical evidence to strengthen our arguments on concern #4. We thereby would like to supplement the actual time costs measured for different LMs in comparison to DPD on a V100 GPU.
> > >
> > > | Component | Time for generating 10s music (s) |
> > > | -------- | :-: |
> > > | Semantic LM | 3.5 |
> > > | Coarse Acoustic LM | 26.4 |
> > > | Fine Acoustic LM | 67.7 |
> > > | **DPD** (5 steps) | **1.3** |
> > > | **DPD** (10 steps) | **2.3** |
> > > | **DPD** (20 steps) | **4.2** |
> > >
> > > *P.S.* The time costs of LMs were measured with a popular unofficial implementation of MusicLM [1] (using 12 RVQ quantizers -- 4 for coarse and 8 for fine acoustic LMs, respectively).
> > >
> > > In other words, adopting DPD in place of coarse and fine LMs can speed up generation roughly 22x to 63x. We will also add these results in our final paper.
> > >
> > > To address the reviewer's other concerns, we will also try to experiment on the "teaching forcing" strategy for training and adding more analysis on VAE-GAN. Yet, because of the tight schedule, please pardon us if these results could not be released during this discussion phase.
> > >
> > > [1] https://github.com/lucidrains/musiclm-pytorch.

---

### Official Review · Reviewer_cbjn · 2023-07-06

**Soundness:** 2 fair
**Presentation:** 2 fair
**Contribution:** 3 good
**Rating:** 4
**Confidence:** 3

**Summary:**

The authors introduce a framework (MeLoDy) for music generation. This new framework significantly improves the inference speed of MusicLM while producing  high-quality results. in terms of musicality, audio quality, and text correlation. The main reason for this speedup is that the authors replaced the coarse and fine acoustic stages in MusicLM and AudioLM from a language model to a dual-path diffusion model (DPD) in the latent space,  while keeping a LM to perform the high level semantic modeling process similar to MusicLM. The authors show that the resulting model can perform similar to MusicLM while being faster at sampling.

**Strengths:**

1. The authors address efficiency, which is a very important issue and can be easily overlooked in new research.
2. I find the idea of combining LM and diffusion models very intuitive and has a lot of potential.
3. The authors put a lot of effort into evaluating the generated samples by using human feedback.
4. The generated samples sound good (since they are not cherrypicked).

**Weaknesses:**

1. Complex pipeline system: The authors propose a complex pipeline with many components, moving parts, and many hyperparameters. This makes the system hard to reproduce, and the contribution of individual components is not clear. The authors provide a limited ablation study in the appendix. I think in a system like this, an extensive ablation study is needed. Examples: (a) What role does the encoder play? (b) Would different encoders affect the quality? (c) The authors used different Mulan audio and text towers compared to MusicLM; what effect does that have? (d) The authors mention they used attention on the course path and an SRU on the fine path; what role do these play? (e) What is the effect of merging and repeating?
I think a lot of these decisions sound arbitrary making the work read like a technical report.
2. Reproducibility, private dataset: the dataset is very vaguely described, making the proposed system even harder to reproduce and evaluate. The dataset plays a large role in the quality of the generation.
3. Clarity: Section 4 in particular is very hard to read; the description of the system is not very clear. Additionally, some details are omitted, which can be in part because of the complexity of the pipeline. I recommend a restructure of this section. The appendix significantly helps in understanding the details.
4. Evaluation: I think a comparison with the more specialized models can greatly improve the quality of the evaluation. Even if the specialized models (like genre) perform better under specific conditions. I believe it's important to illustrate the gap between these models and those that rely on free-form text generation. I think the current evaluation is limited.
5. I believe the related work section is limited. If possible, I think this section should be extended to include related work and alternatives that can be used in the proposed pipeline.

**Questions:**

A- Would you please address the weekness above?
B- In line 158, you say: "In our experiments, we find that the stability of generation can be significantly improved if we use token-based discrete conditions to control the semantics of the music and let the diffusion model learn the embedding vector for each token itself." For me, it's not clear what you mean here: are the tokens coming from the SSL Wav2Vec2-Conformer? How are you "letting the diffusion model learn the embedding vector for each token" if the embedding vectors are coming from the k-means clusters of the Wav2Vec2-Conformer?

C-Line 183, The notation is not very clear for me; I assume the same MLP is performed in parallel for each token. and this is a different MLP than the one in equation 10? Is this correct? If so, I suggest stating the parameterization to make it more clear.

D-Line 192, In order to reduce complexity, you set k = sqrt(L), which allows you to process long sequences with complexity that grows linearly with the sequence length L (after the self-attention over the coarse path); this is the reason you state in the abstract that "infinitely continuable generation." Is that correct? Does not this come with the limitation that as this number K grows, the coarse path becomes even more coarse (more abstract)? Is this explained somewhere?

E-Line 284, why did you choose to use an outdated VGGish model? Wouldn't the results be more accurate if one used more capable models, like a transformer (in AudioGen) or models that are trained to perform musical tasks, e.g., detect musical instruments?

**Limitations:**

The authors discussed the limitations of their work. One additional thing that comes to mind is copyright in music, which is a general issue for generative models. Is there a way to determine if the model reproduces copyrighted material from its training data?

---

> ### Author Rebuttal · Authors · 2023-08-08
>
> We appreciate the thorough review regarding our study. We provide detailed responses to all reviewer's concerns, as summarized below.
>
> ### Response to concern #1
> The reviewer concerned about the **complex system design** of the proposed MeLoDy with **``many components, moving parts, and many hyperparameters''**. We conceive that the term "many" used here could be arguably subjective without a suitable comparing target for comparison. Therefore, we would like to emphasize the following points:
>
> - (1) Since MeLoDy is proposed for an efficient generation of high-quality music, rationally the best comparing target would be the SOTA method/system proposed for music generation up to the date of this submission, i.e., MusicLM.
>
> - (2) Having confirmed a suitable comparing target, we now compare MusicLM against MeLoDy in terms of the number of components:
>   - **MusicLM**: 3 LMs (semantic, coarse & fine), a SSL module (w2v-BERT), a prompt encoder (MuLan) and an autoencoder (SoundStream). In total, there are **6 components** for training MusicLM.
>   - **MeLoDy**: 1 LM, 1 diffusion model (DPD), a SSL module (Wav2Vec2-Conformer), a prompt encoder (MuLan) and an autoencoder (VAE-GAN). In total, MeLoDy consists of **5 components**, which is less than that in MusicLM.
>
> - (3) In fact, we agree with the reviewer on the direction of making less complex music generation system for the ease of reproduction and analysis of individual components. However, it remains hypothetical whether a simple system can generate comparable or even better music audios without particular designs from different perspectives of understanding, e.g., semantic and acoustic perspectives.
>
> ---
>
> ### Response to concern #2
> The reviewer also concerned the **contribution of individual components** and the **limitation of ablation studies**.
>
> - Firstly, the reviewer is right that we have experimented over many possible alternative modules for different sub-tasks, e.g., w2v-BERT v.s. Wav2Vec2-Conformer. In general, this can be casted into a searching problem for a best-performing model, which leads to a combinatorial cost. The ideal starting position of searching for us would be to replicate the settings in MusicLM.
> - Similar to many other research works, we used a leave-one-out comparison to greedily select the best module in each of the sub-task with an acceptable cost. It is noteworthy that the cost of training each possible recipe of our generation model is huge, as we used a large-scale training dataset (257k hours).
> - For each of the individual modules, we investigated suitable objective measures for rigorous comparisons, e.g., SI-SNR for diffusion architectures. However, for other modules like SSL, it is very hard to tell which one is better with an objective measure (if there is any, please let us know.). We therefore can only listen to some random samples (10-100 per system) led by different SSL modules (i.e., Wav2Vec2, Wav2Vec2-Conformer, and w2v-BERT), and subjectively vote for a better module. In common practice, this has been a simple yet effective way to select an audio generative model, since no objective metric can compete with human ears in audio quality assessment.
> - Response to **``(a) What role does the encoder play? (b) Would different encoders affect the quality?''**: If we understand correctly, the reviewer refers to the audio encoder in VAE-GAN. In this sense, the role of encoder is to construct a latent feature space that is robust to the generation errors, as discussed in [1]. We have tried a number of different configurations for VAE-GAN, and we can conclude that different encoders would certainly affect the quality. The output rate of the encoder (a.k.a. frequency of the latent sequence) is especially important, as the audio reconstruction from a higher-frequency latent sequence (e.g. 250Hz in MeLoDy) appears to be much better than a lower-frequency one (e.g. 50Hz in MusicLM). We will add this justification to our revised paper.
> - Response to **``(c) The authors used different Mulan audio and text towers compared to MusicLM; what effect does that have?''**: Since the code and training data of MuLan is not publicly available, the best we can do was to train the model from scratch following the network choices stated in their papers (AST for audio tower; BERT for text tower). We admit that the MuLan we trained on our data cannot be aligned with the one used in MusicLM. Yet, we actually used comparable network architectures for audio and text towers.
> - Response to **``(d) The authors mention they used attention on the course path and an SRU on the fine path; what role do these play?''**: If we understand the question correctly, the reviewer is confused about the design choice of applying attention and SRU for coarse-path and fine-path processing, respectively. In the main paper, we linked the de-noising task in diffusion to the separation problem. We referred to the conclusion in a separation work [2], where the authors found two consistent patterns: (1) in local modeling, the recurrent model performed better than the attention model; (2) in global modeling, the attention model performed better than the recurrent model.
> - Response to **``(e) What is the effect of merging and repeating?''**: Segment merging and repeating operations are originated from [3], where the authors stated two benefits: (1) Dynamic segment scale (termed as multi-granularity in [3]), and (2) speed up coarse-path processing. We will add more discussion on these effects in the revised paper.
>
> [1] R. Rombach, et al. High resolution image synthesis with latent diffusion models. In CVPR, pages 10684–10695, 2022.
>
> [2] M. WY Lam, et al. Effective low-cost time-domain audio separation using globally attentive locally recurrent networks. In 2021 IEEE SLT, pages 801–808. IEEE, 2021.
>
> [3] M. WY Lam, et al. Sandglasset: A light multi-granularity self-attentive network for time-domain speech separation. In ICASSP 2021, pages 5759–5763. IEEE, 2021.

---

> > ### Comment · Reviewer_cbjn · 2023-08-17
> >
> > I'd like to thank the authors for the reply.
> >
> > First, of course, I'd like to agree with the authors that the complexity of the method is subjective. I'd like to stress what I mean for clarity: I think presenting a complex system without an ablation study that shows the contributions of the components, the design choices, and the hyper-parameters is only acceptable in the context of a technical report. I think for a conference paper, the reader should be able to (1) reproduce your work and (2) understand the effect of each component you propose on the results.
> >
> > >  which leads to a combinatorial cost.
> >
> > I agree that trying different combinations is, of course, not feasible, but as you explain, leave-one-out comparisons are standard in the literature. However, I didn't seem to find the leave-one-out comparison in the paper or the appendices.
> > For example, you explain that you started with the musicLM framework and switched from w2v-BERT to Wav2Vec2-Conformer but there are no results showing the effect of this change. Another example, replacing the components of mulan. There, you can also show the role this plays in your results. Another examples include augmenting the text for training mulan etc...

---

> > > ### Author Response · Authors · 2023-08-18
> > > **Response to the reviewer**
> > >
> > > Thanks for the reviewer's insightful comments.
> > >
> > > We would like to reply point by point here:
> > > - (1) **Reproducibility**: We agree that the reproduction of MeLoDy is essential for this paper, therefore we made much effort to formally describe every details of our proposed method (mainly on the design of dual-path diffusion). We believe that our detailed presentation as well as the setting of all hyper-parameters would facilitate the reproduction of this model and our results.
> > > - (2) **Understanding the effect of each component**: We would like to express our regret that not all the design choices have been explained well for their motivations, as we tried to present all the details for better reproducibility. As discussed with other reviewers, we have accordingly modified the structure of our paper, such that more explanations in textual form has been added in the main paper and much detail has been moved to the Appendix. The author rebuttals to the reviewers also include a part of description explaining the design choice of our model. For example, reviewer **ieEr** replied that our further explanations have adequately addressed all his/her concerns.
> > > - (3) **Switching from w2v-BERT to Wav2Vec2-Conformer**: As mentioned in the rebuttal, considering the scale of data we are training and the difficulty of measuring the effects of SSL module, "we can only listen to some random samples (10-100 per system) led by different SSL modules (i.e., Wav2Vec2, Wav2Vec2-Conformer, and w2v-BERT), and subjectively vote for a better module." We will also add this justification in our revised paper. We believe objectively evaluating the SSL module in music/audio generation is still an unsolved open research problem, and is beyond the scope of this paper. If the reviewer has a better suggestion on this evaluation, please let us know.
> > > - (4) **Replacing the components of mulan**: As per mentioned in the rebuttal, we would like to emphasize that "we actually used comparable network architectures for audio and text towers". Please correct us if we misunderstood the meaning of "replacing the components of mulan" stated by the reviewer.
> > > - (5) **``Another examples include augmenting the text for training mulan''**: We agree to the reviewer on this point that the effects of text augmentation on MuLan were not well discussed in the main paper. We would like to express our apology and justify the training pipeline of our MuLan in the revised paper. In fact, after the text augmentation, some objective metrics used in music retrieval, e.g., mAP, were declined yet the music generation on long-form text prompt was obviously improved (improvements were subjectively determined by music professionals). As a result, we finally take the subjective test as the golden rule in evaluation, and ignored somewhat contradictory objective measures.
> > > - (6) **``I didn't seem to find the leave-one-out comparison in the paper or the appendices.''**:  We would like to re-claim our results of leave-one-out comparisons here. In the main paper, since we highlighted two novelties of this paper, each of which has been compared to the conventional methods in our experiments:
> > >   1. The novel **dual-path architecture** for the latent diffusion model (resulting in dual-path diffusion (DPD)): the dual-path architecture has been compared to UNet-1d [1] and UNet-2d [2]:
> > >
> > > | Architecture    | Velocity MSE (↓) | SI-SNR (↑) |
> > > | -------- | ------- | ------- |
> > > | UNet-1d [1]  | 0.13   | 5.33    |
> > > | UNet-2d [2] |  0.15  |  4.96  |
> > > | **DPD** (Ours)  | **0.12**    | **6.15**   |
> > >
> > > 2. The novel **angle schedule** for angular-parameterized diffusion:
> > >
> > > | Angle Schedule    | Steps | FAD (↓)  | MCC (↑)  |
> > > | -------- | ------- | ------- |------- |
> > > | Uniform [1]  | 10 | 8.52 | 0.45 |
> > > | Uniform [1]   | 10 | 6.31 | 0.49 |
> > > | Ours proposed in Eq. (4)  | 10 | **5.93** | **0.52** |
> > > |  Ours proposed in Eq. (4)  | 20 | **5.41** | **0.53** |
> > >
> > > Despite an argument of "technical report", we sincerely ask the reviewer to re-consider the significance of this work, as a novel music generation model that first allow generating music audio of high quality and competitive musicality faster than real-time on a consumer-level GPU. We believe our revised paper has been greatly improved with the proposed amendment in the rebuttal and has addressed all concerns about the justification of the design choices.
> > >
> > > [1] Flavio Schneider et al.. Moûsai: Text-to-music generation with long-context latent diffusion. 2023.
> > >
> > > [2] S Forsgren and H Martiros. Riffusion - stable diffusion for real-time music generation. 2023.

---

### Official Review · Reviewer_ieEr · 2023-07-14

**Soundness:** 3 good
**Presentation:** 3 good
**Contribution:** 3 good
**Rating:** 7
**Confidence:** 4

**Summary:**

This paper proposes MeLoDy, a diffusion-based text-to-music generation system. The system comprises four parts: 1) the MuLan contrastive model, 2) semantic language modeling using Wav2Vec2-Conformer, 3) a dual-path diffusion model to generate the latent, and 4) a 1D VAE-GAN autoencoder for learning the latent. The greatest contribution of this paper is the dual-path diffusion model, while other components follow a similar paradigm to that proposed in previous models, albeit with minor upgrades.

In the dual-path diffusion model, instead of applying diffusion on the full audio length, the paper suggests segmenting the target of length L into approximately sqrt(L) chunks. A RoFormer model is then used to process inter-segment, while an RNN is utilized to process the intra-segment. The dual-path diffusion model trains on a velocity objective with a novel linear angle schedule.

A subjective listening test with music producers as listeners demonstrates that the proposed model outperforms previous methods in quality but falls short in musicality and text correlation.

**Strengths:**

The paper proposes a novel dual-path diffusion model, which could be an effective alternative to the current design of the diffusion model. A new schedule for the diffusion process is also proposed. The model generates satisfactory results, as shown on the demo website, and the writing in the paper is clear.

**Weaknesses:**

From my perspective, there are two minor weaknesses:

1. The authors should further justify the choice of the dual-path diffusion model in terms of its effectiveness. When compared to an alternative diffusion module for the proposed system, the prediction length in the dual-path diffusion is smaller. However, according to the paper, in the dual-path diffusion, at each diffusion (reverse) step, the input needs to run through two modules: RoFormer and SRU. While I can imagine that the RoFormer can be run in parallel for both inter-segment and intra-segment, I believe the SRU, as a type of RNN, still requires sequential computation. Therefore, an alternative diffusion model that works on 1D (waveform) or 2D (spectrogram) latent could leverage the benefits of parallel computing and run faster than the dual-path diffusion model on a GPU under the same sampling steps. I would appreciate the authors' insights on this point. It would be beneficial if this could be included in the paper to provide a more solid justification for the design choice.

2. The experiments show that the proposed model does not outperform in terms of musicality and text correlation, which calls the design of the proposed model into question. However, I acknowledge that the data used in each model differ, making a controlled and reproducible comparison nearly impossible.



**Questions:**

Typos:
Line 277: The sentence appears to be incomplete: "For sampling, the predicted velocity is linearly combined as ."

Issues with introducing the system:
In section 4.1.1, the concept of "velocity" should be introduced beforehand. Its sudden appearance made my initial reading confusing.

Also, I would like to know about the justification of the dual-path diffusion design choice. Please see the first point in the weakness section.

**Limitations:**

The limitations of the proposed system are adequately discussed in the final section.

---

> ### Author Rebuttal · Authors · 2023-08-09
>
> We are grateful for the reviewer's overall positive response. The concerns and questions raised by the reviewer is addressed below.
>
> ### Response to concern #1
>
> **Justification of the choice of the dual-path diffusion model**: Thanks for the insightful comments. We would like to first clarify one fact about dual-path network used in DPD. Suppose the input to the diffusion model is $\mathbf{z}\_\text{noisy}\in\mathbb{R}^L$. After segmentation, there are two shortened sequences to be processed by the DPD blocks:
> - **Intra-segment sequences** (fine path): Each intra-segment sequence has a length of $K\approx \sqrt{L}$. The processing is parallelized across $S=\lceil{\frac{2L}{K}}\rceil+1$ segments. In practice, suppose the input to DPD block has a shape of $[B, D, S, K]$, we can easily implement this parallelization in PyTorch by `.permute(0, 2, 3, 1).reshape(B * S, K, D)`, where $B$ is the batch size and $D$ is the hidden dimension. When using SRU, the computation cost is $\mathcal{O}((BS)(D^2K))=\mathcal{O}((B\sqrt{L})(D^2\sqrt{L}))$.
> - **Inter-segment sequences** (coarse path): On the other hand, each intra-segment sequence has a length of $S$. The processing is parallelized across $K$ frames (or merged frames). Let's first ignore the case of segment merging and repeating. The parallelization can be similarly implemented in PyTorch by `.permute(0, 3, 2, 1).reshape(B * K, S, D)`. When using attention-based sequence processing method, the computation cost is $\mathcal{O}((BK)(D^2K+DS^2))=\mathcal{O}((B\sqrt{L})(D^2\sqrt{L}+DL))$.
>
> As mentioned by the reviewer, since the batched computation is specifically speeded up in GPU (with batch sizes of $BS$ and $BK$, respectively, for the above two sequences), we separate the considerations of batch axis and other axes in Big-O notation. On the other hand, considering the ``alternative diffusion model that works on 1D (waveform) or 2D (spectrogram) latent'' mentioned by the reviewer, we note that, without segmentation, an attention-based network operating on the full input sequence of length $L$ gives rise to a cost of $\mathcal{O}((B)(D^2L+DL^2))$. In this sense, we can show in Big-O notation that DPD should be faster than any attention-based network operating on the full sequence of length $L$:
> - When $D$ is much larger than $L$, the dominant term in the attention network of interest would be $\mathcal{O}((B)(D^2L))$. The dominant term in DPD would be $\mathcal{O}((B\sqrt{L})(D^2\sqrt{L}))$, caused by SRU. Since the batch axis in DPD can processed much faster with GPU, we conclude that DPD is faster than than the attention network of interest.
> - When $L$ is much larger than $D$, the dominant term in the attention network of interest would be $\mathcal{O}((B)(DL^2))$. The dominant term in DPD would be $\mathcal{O}((B\sqrt{L})(DL))$, caused by RoFormer. It is obvious that, even not considering the GPU acceleration on the batch axis, the cost of DPD in $\mathcal{O}(BDL^{3/2})$ is still lower than that in the attention network of interest $\mathcal{O}(BDL^2)$.
>
> We also reach to a consistent conclusion in practice. During our ablation on different architectures, we also validated that the DPD model is faster than alternative diffusion models (i.e., 1D-UNet and 2D-UNet with cross-attention) for sampling one step, when both networks are constrained with the same model size.
>
> Regarding the particular choice of SRU and Roformer in DPD, we refer to the linkage between the de-noising task in diffusion to the separation problem, as stated in the main paper. In the context of separation, we accord to the findings in a separation work [2]: (1) in local modeling, the recurrent model performed better than the attention model; (2) in global modeling, the attention model performed better than the recurrent model. Following from these conclusions, we experimented through different RNNs (i.e., GRU, LSTM, and SRU) and attention networks (i.e., Transformer and Roformer), and eventually selected overall the best combinations (SRU+Roformer).
>
> ---
>
> ### Response to concern #2
>
> **``the proposed model does not outperform in terms of musicality and text correlation''**: First, we would like to note that the baseline samples we used for comparisons are (at least partially) cherrypicked, as stated in the main paper of Noise2Music. In contrast, we used the non-cherrypicked ones for all the text prompts taken from MusicLM and Noise2Music. In fact, the number of samples released on our demo page is more than 300, which prohibits the chance of cherrypicking sample for each prompt. Presumably, the sample variances of diffusion and LM could cover marginal differences in musicality and text correlation. In other words, from a practical sense, if we need to improve the musicality and text correlation given a text prompt, we can re-sample with the same prompt until we get a desired sample. Whereas, the generation speed and audio quality (denoting upper bound of the generation quality) aspects are rather steady -- they would not drastically vary even if we draw for more samples.

---

> > ### Comment · Reviewer_ieEr · 2023-08-10
> >
> > Thanks, authors, for explaining. According to the reply and the reply to other reviewers, I believe my concerns have been addressed.

---

### Official Review · Reviewer_hBX7 · 2023-07-26

**Soundness:** 3 good
**Presentation:** 2 fair
**Contribution:** 3 good
**Rating:** 6
**Confidence:** 3

**Summary:**

The paper introduces MeLoDy, a music generation method combining Language Models (LMs) and Diffusion Probabilistic Models (DPMs). It addresses the challenge of generating music from diverse free-form text descriptions while mitigating high computational costs associated with current state-of-the-art models like MusicLM and Noise2Music. Key contributions include the development of MeLoDy, which significantly reduces computational requirements, the proposal of efficient Dual-Path Diffusion (DPD) models, an improved sampling scheme for DPD, and a successful implementation of audio VAE-GAN for effective continuous latent representation learning.

**Strengths:**

The DPD model is particularly innovative as it is a variant of continuous-time diffusion probabilistic models that operates on low-dimensional latent representations.

As for significance, the paper offers a substantial contribution to the field of music generation by showcasing how complex relationships across long-term contexts can be modeled and by overcoming limitations of prior works.

**Weaknesses:**

The proposed method builds upon existing research, employing the diffusion probabilistic model for music generation. Although the combination with the dual-path architecture is intriguing, it still largely borrows from previous architectures like AudioLM, MusicLM, and MuLan.

**Questions:**

What's the rule of thumbs for selecting the number of chunks when training?

**Limitations:**

The authors have discussed about the limitations of this work in the discussion section.

---

> ### Author Rebuttal · Authors · 2023-08-09
>
> We are grateful for the reviewer's an overall positive response. The question raised by the reviewer is addressed below.
>
> **Question:** What's the rule of thumbs for selecting the number of chunks when training?
>
> **Answer:** If we understand correctly, by "the number of chunks", the reviewer refers to the hyperparameter $M$, which defines the number of chunks for dividing the diffusion input with different noise scales. To understand the rule of thumbs for determining $M$, we first would like to re-visit the functionalities of using multi-chunk input. In essence, introducing such a input chunking technique leads to ``infinitely continuable generation'' for DPD. Recall that the network $v_\theta(\mathbf{z}\_{\text{noisy}}; \mathbf{c})$ is conditional on with a noise scale vector ($\boldsymbol\delta\in\mathbb{R}^{L}$) that records the time-aligned noise scales of all elements in $\mathbf{z}\_{\text{noisy}}\in\mathbb{R}^{L}$. Because of this time-aligned condition, the model is capable of de-noising partial regions of the noisy input.
>
> For example, given a model trained on 10s of music audio and let M = 4, at the first run we can generate 10s sample. To extend for 2.5 more seconds, we can drop the first chunk of the generated sample (7.5s left) and append a new chunk of 2.5s white noise to the end of the sample. The resulted 10s is passed to the diffusion model as the input. Then, taking advantage of the input chunking scheme we used for training, we can intentionally set the first $\frac{3L}{4}$ values in $\boldsymbol\delta$ to zeros and set the last $\frac{L}{4}$ values in $\boldsymbol\delta$ to one. In this way, the model is capable of ignoring the first 3 chunks and only de-noises the the last appended chunk. At the same time, the de-noising process of the last chunk (2.5s) would depend on the first previously generated 7.5s for audio continuation.
>
> By concerning the effect of $M$ on generation, we can select the number of chunks by considering 4 rules:
> - (1) **The length of training input**: MeLoDy was trained with 10s music clips, therefore it is better to select an $M$ such that $L$ is divisible by $M$. In our setting, 10s of music audio leads to $L=2500$ (250Hz * 10s). Setting $M=4$ leads to a length of $\frac{L}{M}=625$ latents in each chunk.
> - (2) **The length of segment size in DPD**: In order to perform segmentation in DPD, we should ensure $\frac{L}{M} \geq K$. In our setting, $K=80$, therefore the constraint is satisfied.
> - (3) **The speed of continuation**: A larger $M$ leads to a smaller chunk $\frac{L}{M}$, and the continuation would become slower.
> - (4) **The quality of continuation**: While different $M$ values correspond to different length of historical contents that assist the de-noising of the last chunk, when we set a smaller $M$ the length of previously generated contents used also becomes shorter.
>
> From rule (1), we have the possible values $M\in\set{1, 2, 4, 5, 10, 20, 25, 50, 100, 125, 250, 500, 625, 1250, 2500}$ (but $M=1$ is a dummy case that does not support continuation). From rule (2), we can further narrow the value set to $M\in\set{1, 2, 4, 5, 10, 20, 25}$. From rule (3), we conceive that it is more practically feasible to extend one-fifth to one-fourth of the generated audio at each run of the de-noising process. From rule (4), we experimented both $M=4$ and $M=5$ and assess their quality in continuation. We empirically found subtle differences in quality when using $M=4$ and $M=5$, therefore we finally opted for $M=4$ to enjoy a faster continuation.

---

### Decision · Program_Chairs · 2023-09-21

**Decision:**

Accept (poster)

**Comment:**

This paper introduces a text-to-music latent diffusion model, that conditions on LM generated semantic tokens, with a target latent space obtained from an adversarial VAE. To speed up decoding, the paper proposes a dual-path architecture for diffusion in the latent space which is then decoded using the audio VAE-GAN.

The overall score leans positive. All of the reviewers found the proposed method to be innovative and significant, and that it addresses the important issue of efficiency. Clarification of how the model enables improved efficiency was needed, and the authors have addressed most of the reviewers' questions in the rebuttal.

A couple reviewers raised the need for ablation studies, as the method requires many components and design decisions. The authors have explained some of their choices and how they tested for various modeling alternatives. We encourage the authors to include those results in the final paper.

The paper demonstrated improvement on metrics of audio quality, while some reviewers were concerned that the method performed worse on other objective and subjective metrics. The authors provided a reasonable explanation, and a few reviewers did find the samples from the demo page compelling.

Overall, this paper presents a novel and significant contribution that is well-executed. It offers a novel example of combining the strengths of LM and diffusion-based models, enabling  improved efficiency. Hence, the recommendation is to accept the paper.